# A cage-on-MOF strategy to coordinatively functionalize mesoporous MOFs for manipulating selectivity in adsorption and catalysis

Yu Liang[1,2,5], Xiaoxin Yang[1,3,5], Xiaoyu Wang[4], Zong-Jie Guan[1,2], Hang Xing ◉[1,3] ✉ & Yu Fang ◉[1,2] ✉

Functionalizing porous materials with capping agents generates hybrid materials with enhanced properties, while the challenge is how to improve the selectivity and maintain the porosity of the parent framework. Herein, we developed a "Cage-on-MOF" strategy to tune the recognition and catalytic properties of MOFs without impairing their porosity. Two types of porous coordination cages (PCCs) of opposite charges containing secondary binding groups were developed to coordinatively functionalize two distinct porous MOFs, namely MOF@PCC nanocomposites. We demonstrated that the surface-capped PCCs can act as "modulators" to effectively tune the surface charge, stability, and adsorption behavior of different host MOF particles. More importantly, the MOF@PCCs can serve as selective heterogeneous catalysts for condensation reactions to achieve reversed product selectivity and excellent recyclability. This work sets the foundation for using molecular cages as porous surface-capping agents to functionalize and manipulate another porous material, without affecting the intrinsic properties of the parent framework.

As an emerging class of crystalline porous material, metal-organic framework (MOF) composed of modular metal nodes and organic ligands has demonstrated its potential for gas storage, molecular separation, and heterogeneous catalysis[1–14]. Particularly, MOFs with large cavities are becoming topics of considerable interest due to their large open channels which enable the rapid transport of guest molecules and the encapsulation of macromolecules[15–19]. However, unmodified MOFs are reported to possess potential selectivity issues against substrate molecules with similar sizes, leading to undesired side-products as well as affected separation performance (Fig. 1a)[20]. To realize the full potential of porous MOF materials to function in various scenarios for practical applications, it is important to equip porous MOFs with selectivity towards different substrates of similar chemical or physical properties. Common strategies often involve bottom-up synthesis or post-synthetic modification to create a substrate-specific accessible void within the framework, generating substrate-selective porous MOFs[21–23]. Thus far, different types of surface capping agents have been successfully incorporated onto the external surface of various MOFs, including polymers, phospholipids, phenylsilanes, and biomacromolecules such as proteins and oligonucleotides, generating hybrid materials with improved properties[24–30]. For example, the Janiak group applied MIL-101 to encapsulate cucurbit[6]uril (CB6), obtaining MOF-CB6 composites with enhanced $CO_2$ adsorption and gas separation performance[27]. Li et al. coated MOF particle surface with different organic macrocycles and polymers, and the modified MOFs showed improved solvent dispersity and colloidal stability[28]. Despite these achievements, most approaches reported so far use nonporous capping agents to

[1]State Key Laboratory of Chemo/Bio-Sensing and Chemometrics, College of Chemistry and Chemical Engineering, Hunan University, Changsha 410082 Hunan, China. [2]Innovation Institute of Industrial Design and Machine Intelligence Quanzhou-Hunan University, Quanzhou 362801 Fujian, China. [3]Institute of Chemical Biology and Nanomedicine, Hunan University, Changsha 410082 Hunan, China. [4]Kuang Yaming Honors School, Nanjing University, Nanjing 210023, China. [5]These authors contributed equally: Yu Liang, Xiaoxin Yang. ✉e-mail: hangxing@hnu.edu.cn; yu.fang@hnu.edu.cn

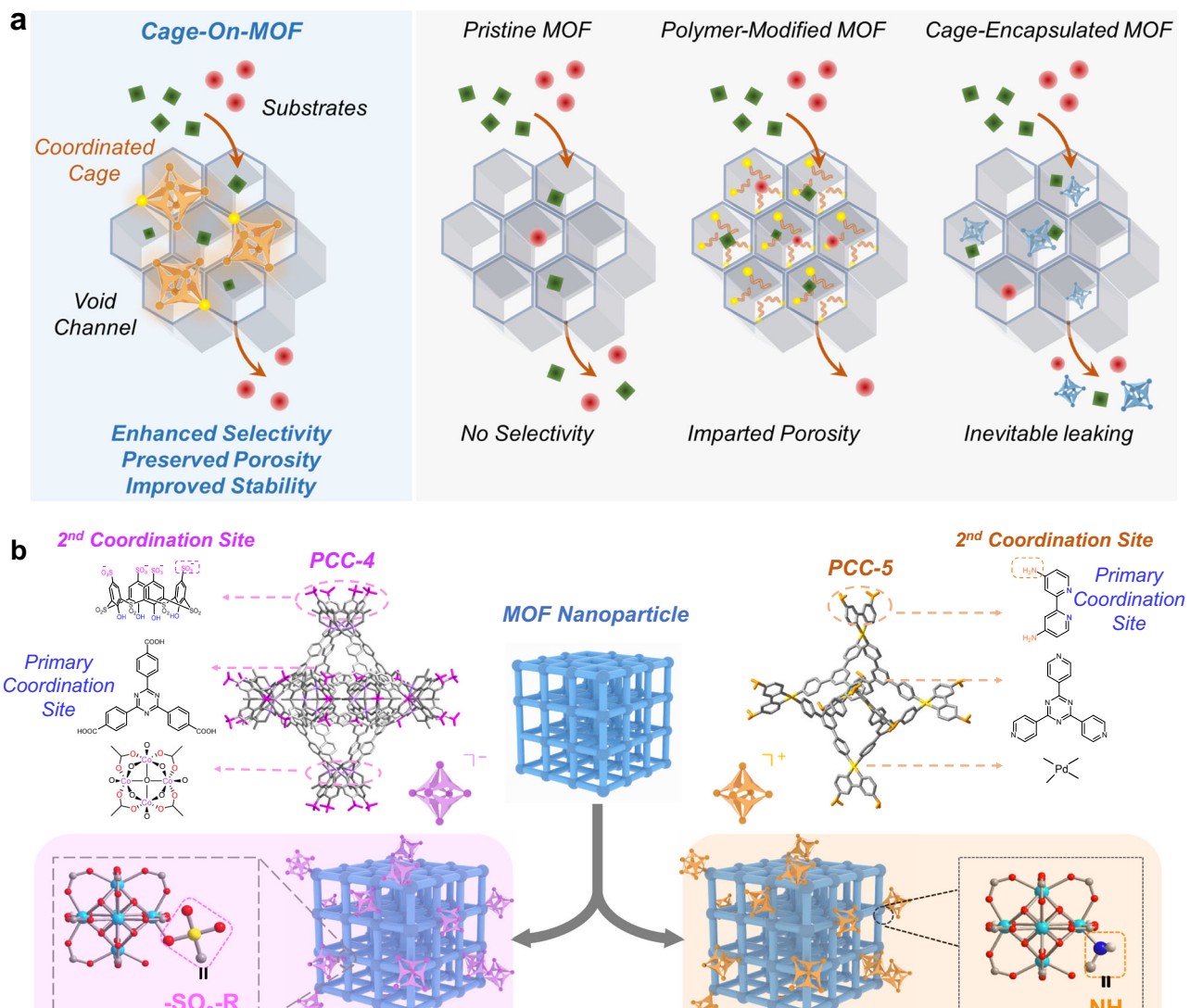

**Fig. 1 | Illustration of the "Cage-on-MOF" strategy. a** Advantages of the "Cage-on-MOF" material over pristine MOF, polymer-modified MOF, and cage-encapsulated MOF. **b** X-ray single crystal structure of PCC-4 (Supplementary data 1) and PCC-5 with secondary coordination sites. The cartoon for showing the cage-coordinated MOF composites: MOF@PCC-4 and MOF@PCC-5.

modify the exterior surface, thereby inevitably affecting the porosity and related proprieties of the host framework (Fig. 1a).

In order to enhance the characteristic properties of MOFs and introduce additional functionalities through surface modification without compromising the structural porosity, it is crucial to design an optimal surface capping agent with inherent porosity for facile guest molecule transport and adjustable functions for precise recognition. In this vein, porous coordination cages (PCCs), also known as metal-organic polyhedrons (MOPs) or cages (MOCs), as a class of discrete supramolecular cage structures of 1–20 nm size with intrinsic cavities attract our attention[31–37]. PCCs have proven to be useful in the context of host–guest chemistry, catalysis, and region-selective organic reactions[31–37]. Owing to their intrinsic porosity and highly tunable properties, such as charges, sizes, and functional groups, PCCs can potentially serve as ideal surface-capping agents to modify the exterior surface of MOF particles, forming multicomponent porous materials with enhanced functionalities and preserved accessible voids. To date, a limited number of studies have explored the use of molecular cages as functionalization agents for MOFs, but these investigations have only demonstrated the successful encapsulation of molecular cages

within the cavities of the host framework[38–40]. For example, Li et al. reported the encapsulation of the $Pd_6L_4$ cage within the cavity of MIL-101, which showed improved stability in the oxidation of benzyl alcohol to benzaldehyde[38]. Nevertheless, most of the "Cage-MOF" hybrid materials reported so far have been generated through non-covalent encapsulation, wherein small cages are placed inside the host MOF cavities, potentially compromising the porosity of MOFs due to the haphazard arrangement of cages within the host voids. Furthermore, non-covalent interactions may also pose a concern regarding the stability of the "Cage-MOF" hybrid material in extreme conditions (Fig. 1a)[39,40].

To realize target-selective adsorption and catalytic reaction using MOFs, herein, we present a "Cage-on-MOF" strategy to homogeneously incorporate PCCs onto the external surface of MOF particles through the coordination with the surface-exposed metal sites (Fig. 1a). Two PCCs have been designed and synthesized as surface-capping agents, defined as **PCC-4** (Supplementary data 1) and **PCC-5**, that are bulky in size and have different secondary coordination sites with different charges (-SO$_3^-$ and -NH$_2$) (Fig. 1b). We selected PCN-222 and MIL-101, two iconic MOFs known for their large surface area and

accessible voids, as the model structures[16,18]. By incorporating PCCs with secondary coordination sites onto these MOFs, the exposed sulfate and amino groups on PCC can stably bind with metal sites exposed on the external surface of MOF particles[41–47], forming four distinct porous materials, namely "MOF@PCC" (Fig. 1b). Because PCCs are highly charged, the negative PCC-4 dramatically can reverse the surface charge of MOF@PCC-4 to negative, while the positive PCC-5 further increases the positive charge of MOF@PCC-5. Importantly, we demonstrated that porous PCCs are coordinated primarily on the exterior surfaces of both PCN-222 and MIL-101 without compromising their porosity due to their steric hindrance and charge repulsion. Thus, the modified MOFs exhibited similar gas adsorption capacities as the pristine MOFs. The MOF@PCCs also exhibited improved photo- and chemical stability under various harsh conditions. More importantly, by capping the surface of MOFs with porous cages of different charges, the adsorption preference and the catalytic selectivity of MOF@PCCs can be deliberately tailored, which in turn results in the enhanced substrate- and product-selectivity for dye adsorption and condensation reactions, demonstrating the effectiveness of the "Cage-on-MOF" strategy.

## Results and discussion
### Synthesis of PCCs and MOF@PCCs

Previously, our group reported two PCCs with 6- negative charge (PCC-2b) and 12+ positive charge (PCC-3), which showed superior guest inclusion and catalytic properties[48,49]. However, these two PCCs do not contain secondary coordination moieties and therefore can hardly be used as capping agents to functionalize other materials due to the lack of binding sites. To develop PCC-based surface-capping agents for functionalizing MOFs, we designed and synthesized PCC-4 and PCC-5 with two distinct secondary coordination groups respectively: sulfate (-SO$_3^-$) for PCC-4 and amino (-NH$_2$) for PCC-5 (see Methods and Supporting Information for synthesis details). Single-crystal X-ray diffraction (SC-XRD) (Supplementary Figs. 1, 2, Supplementary Table 1, Supplementary data 1), $^1$H NMR (Supplementary Fig. 3), and electrospray ionization-mass spectrometry (ESI-MS) were performed to identify the molecular structures of these two PCCs, which showed the same topologies to reported PCC-2b and PCC-3 respectively (Supplementary Figs. 4, 5 and Supplementary Table 2). Specifically, PCC-4 consists of Co$_4$-$\mu_4$-OH clusters with exposed sulfate groups, while PCC-5 consists of Pd (II) nodes with exposed amino groups (see Supporting Information Section 2 for structure details). Since sulfate and amino groups were reported with the ability to coordinate with metal clusters[41–47], both PCCs can be immobilized onto host mesoporous MOF surface via stable coordination bonds (Structure details see Methods and Supporting Information Section 2). For comparison, we also prepared PCC-2b and PCC-3 without secondary coordination sites to serve as control agents using our previously reported approaches[48,49]. Their corresponding molecular structures were identified using SC-XRD (Supplementary Figs. 6, 7).

For our initial study, PCN-222, composed of Zr$_6$($\mu_3$-OH)$_8$(OH)$_8$ cluster and tetrakis(4-carboxyphenyl)porphyrin ligand, was chosen as the model mesoporous framework. Nanoscale PCN-222 particles were synthesized as uniform shuttle-like nanocrystals with an average size of *ca.* 200 nm in length and *ca.* 50 nm in diameter, as demonstrated by transmission electron microscopy (TEM) and scanning electron microscopy (SEM) characterizations (Fig. 2 and Supplementary Figs. 8, 9)[50]. To incorporate PCCs onto MOFs via coordination, excess PCCs were mixed with as-synthesized PCN-222 nanoparticles in DMF or MeCN and stirred at 50 °C for 12 h. The resulting two PCN-222@PCCs were collected by centrifugation and were washed with DMF and MeCN multiple times to remove unbound cages.

### Characterizations of MOF@PCCs

TEM and high-angle annular dark-field scanning transmission electron microscope (HAADF-STEM) results revealed that both PCN-222@PCC nanoparticles adopted the same morphology and particle size as the pristine MOF (Fig. 2a, b), suggesting the well-maintained structural integrity during the coordination modification. Energy dispersive X-ray spectroscopy (EDX) elemental mapping (Fig. 2b, d) and linear scanning patterns (Supplementary Fig. 10) conducted on the two PCN-222@PCC particles provided clear evidence of the uniform distribution of Co (from PCC-4) and Pd (from PCC-5) elements throughout the entirety of the MOF particle, thereby suggesting the homogeneous immobilization of PCCs onto the host framework. Importantly, a further examination of the Zr/Co and Zr/Pd element ratios, which indicate the proportion of metal clusters on the MOF relative to the cage molecules in the linear scanning profiles from the particle's edge to its center, revealed a distinct volcano-shaped curve. This curve demonstrated a lower Zr/PCCs ratio at the thin edge and a higher Zr/PCC ratio at the thick center, thereby suggesting that the immobilization of PCCs likely occurs on the surface of the framework rather than being embedded within it (Supplementary Fig. 11).

Powder X-ray diffraction (PXRD) measurement proved that both PCN-222@PCCs maintained the same crystal after the coordination modifications (Fig. 2e). Zeta potential results clearly showed a negative charge for PCN-222@PCC-4 and an increased positive charge for PCN-222@PCC-5, confirming the incorporation of PCCs (Fig. 2f). It was found that the negative PCC-4 reversed the surface charge of MOF@PCC-4 from positive to negative −20 mV, while the positive PCC-5 further the positive charge of MOF@PCC-5 to +26 mV. The coordination of PCCs on PCN-222 was further verified by Fourier transform infrared spectroscopy (FT-IR) (Fig. 2g). As shown in Fig. 2g, PCN-222@PCC-4 exhibited a characteristic peak centered at 1100 cm$^{-1}$ which belonged to -S=O (from PCC-4), while PCN-222@PCC-5 showed an 810 cm$^{-1}$ peak which was assigned to pyridyl moiety (from PCC-5). X-ray photoelectron spectroscopy (XPS) was also performed to study the coordination environments of Zr and O in PCN-222@PCCs before and after cage modifications. A downshift of both Zr 3$d_{3/2}$ and Zr 3$d_{5/2}$ binding energy peaks centered at 185 and 183 eV, respectively, was observed after immobilization of PCC-4 and PCC-5 on PCN-222 (Fig. 2h and Supplementary Fig. 12). The recorded downshifted binding energy of Zr (IV) oxidation states after PCC modification is consistent with previous studies, which can be attributed to the increase of electron density of the Zr$_6$ cluster from ligand coordination[41–43]. Furthermore, the C-O strength bond downshifted, indicating additional binding to the secondary coordination sites on PCCs (Supplementary Fig. 12). To evaluate the porosity of PCN-222@PCCs after surface modification, N$_2$ isotherms at 77 K were measured after thermal activation to remove solvent molecules from pores. Both PCN-222@PCCs exhibited almost identical type VI adsorption isotherms with a slightly decreased adsorption capacity at high pressure (5.6%) and a similar BET surface area to pristine MOF, suggesting that the coordination of PCCs did not affect the host porosity (Fig. 2i and Supplementary Fig. 13). Taking into account that PCCs displayed minimal gas adsorption (Supplementary Fig. 14), the presence of two plateaus in the adsorption isotherm curves can be ascribed to the inherent adsorption characteristics of the parent PCN-222 framework[16]. The number of modified cages on the MOF surface was measured using inductively coupled plasma optical emission spectrometry (ICP-OES). The weight percentages of the two PCCs in PCN-222@PCC-4 and PCN-222@PCC-5 were determined to be 5.40 wt% and 1.98 wt%, respectively (Supplementary Table 3). Based on the calculated surface density, the molar ratio of surface-loaded PCCs versus Zr$_6$ clusters in PCN-222@PCC-4 and PCN-222@PCC-5 was identified as 10.50% and 8.83%, respectively (Fig. 2j). It is important to note that despite the relatively low weight percentage loading of PCCs on the MOF, a significant amount of PCCs are immobilized onto the

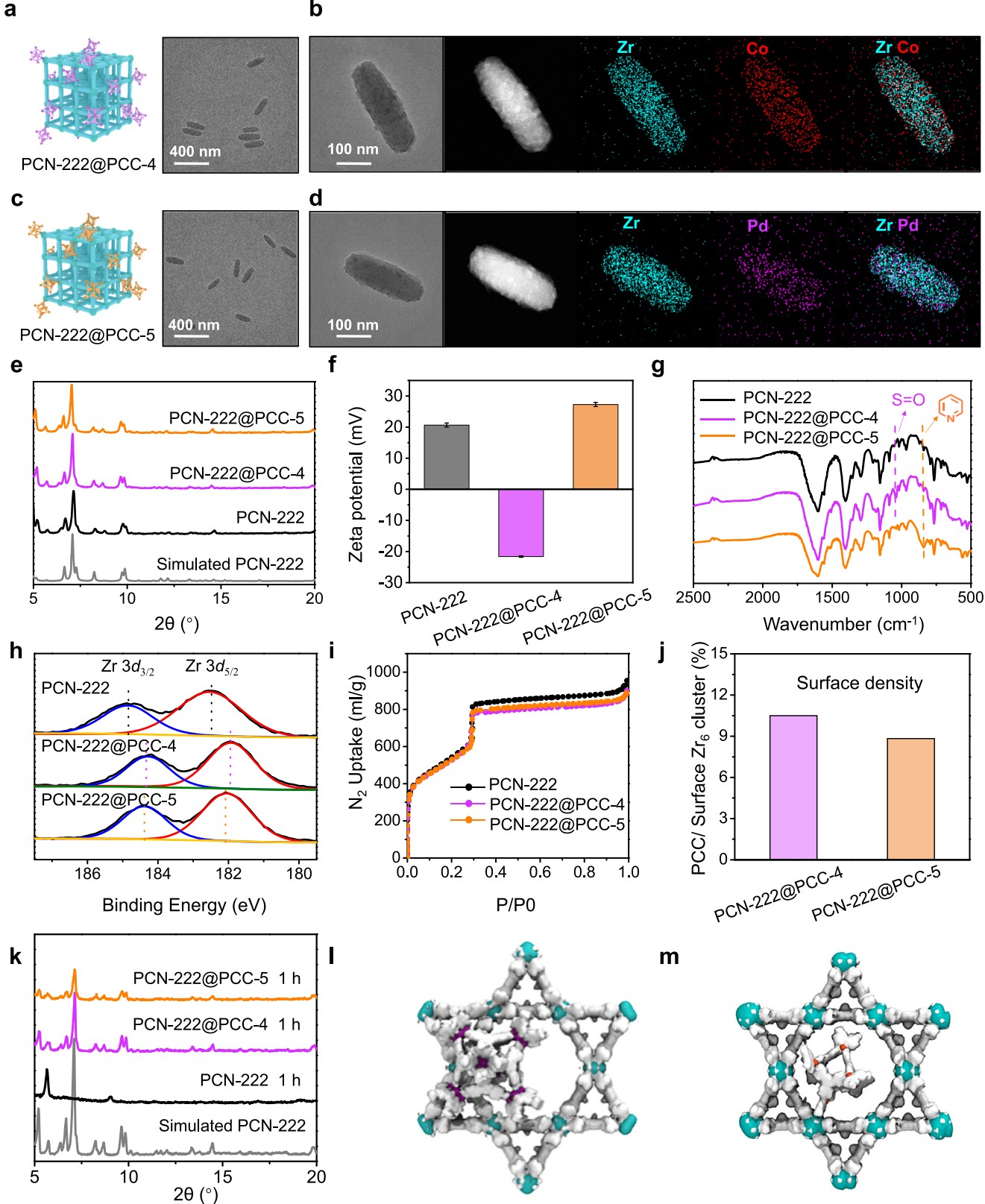

**Fig. 2 | Characterizations of the PCN-222@PCCs.** TEM images of PCN-222@PCC-4 (**a**) and PCN-222@PCC-5 (**c**). STEM image and EDX element mapping of a single particle of PCN-222@PCC-4 (**b**) and PCN-222@PCC-5 (**d**). PXRD patterns (**e**), Zeta potential measurements (Data are presented as the mean ± SD.) (**f**), and FT-IR spectra (**g**) of pristine PCN-222, PCN-222@PCC-4, and PCN-222@PCC-5. The XPS spectra show Zr $3d_{3/2}$ and $3d_{5/2}$ binding energy of PCN-222, PCN-222@PCC-4, and PCN-222 (**h**). $N_2$ adsorption isotherms of PCN-222@PCCs. The $N_2$ uptake was normalized to the same mass of PCN-222 (**i**). The surface density of PCN-222@PCCs (**j**). Photostability of PCN-222@PCCs in benzylamine methanol solution (**k**). Space-filling illustrations generated from molecular dynamics simulations showing the representative structures of PCN-222@PCC-4 (**l**) and PCN-222@PCC-5 (**m**).

surface of MOF particles, largely changing their surface properties. As a result, this alteration of surface properties of MOF can lead to profound changes in various physical and chemical characteristics, including solubility, surface adhesion, stability, gas adsorption, and catalytic properties, which have been reported in previous publications[26,51–54]. Similarly, we observed that **MOF@PCC** composites with PCCs as surface capping agents have been found to exhibit improved chemical and thermal stability (Fig. 2k and Supplementary Figs. 15–18). Furthermore, the binding interaction between the surface PCCs and the parent MOFs is strong enough for further applications, such as reversible dye adsorption and heterogeneous catalysis. Hence, the use of porous molecular cages as surface-capping agents to functionalize a mesoporous MOF can produce a "porous composite", which is in stark difference from the previous modification methods that usually result in the loss of the accessible voids.

The existing location of the cage was studied by molecular dynamics (MD). MD simulations revealed that almost no cages entered pores in a ca. 10 nm-sized lattice system, which proved that PCC-4 and PCC-5 were unable to enter the pores of PCN-222 due to steric hindrances and electrostatic repulsions (Fig. 2l, m, Supporting Information Section 4). By further calculating the number of surface-exposed $Zr_6$ clusters per PCN-222 nanoparticle based on a simplified particle mode and exposed lattice planes, we determined 1 cage molecule occupied in an area of ca. 10 $Zr_6$ clusters. In contrast, **PCC-2b** and **PCC-3** without secondary coordination groups can only be encapsulated within the pores of MOFs (Supplementary Figs. 19–22). As a result, the $N_2$ uptake capacity of modified PCC-222 was significantly reduced by ca. 36% after loading with **PCC-2b** or **PCC-3** (Supplementary Fig. 22), indicating a decrease in the accessible voids. Because of the insufficient binding interactions, the loading amount of molecular cages on

MOF was determined to be only ca. 10.7 mg/g (1.06 wt%) for **PCC-2b** and 8.2 mg/g (0.81 wt%) for **PCC-3**, which were considerably lower than those of their congeners with secondary coordination sites (Supplementary Table 3 and Supplementary Fig. 23).

## Selective adsorption of dye molecules

Considering the surface potential of MOFs can be altered dramatically via PCC modification, we next sought to investigate the use of **PCN-222@PCCs** as a model adsorbent for selective adsorption of dyes with opposite charges in an aqueous solution (Fig. 3a). Positively-charged methylene blue (MB) and negatively-charged Eosin Y (EY) were selected in this study. First, **PCN-222@PCCs** were added to the solution containing a single dye to evaluate the adsorption capacity and kinetics. All MOF-based adsorbents exhibited a concentration-dependent positive correlation with the adsorption capacity (Supplementary Figs. 24–27). For cationic dye MB, with the increase of dye concentration (0–50 mg/L), the maximum adsorption capacity of PCN-222 and **PCN-222@PCCs** increased to reach equilibrium in the range of 125–175 mg/g (Supplementary Fig. 24). Interestingly, compared with the pristine MOF (148 mg/g adsorption capacity), **PCN-222@PCC-4** showed an increased maximum adsorption capacity of 175 mg/g. The enhanced adsorption capacity for the cationic dye could be attributed to the electrostatic attraction between the positively-charged MB and negatively-charged **PCN-222@PCC-4**. In contrast, because of the same charge repulsion between MB and **PCN-222@PCC-5**, the maximum adsorption capacity was decreased to 125 mg/g. The time-resolved dye adsorption kinetics was also recorded from 0 to 120 min. Within the first 5 min, the adsorption rate dramatically elevated, then gradually decreased until reaching the equilibrium (Supplementary Fig. 25). Similarly, **PCN-222@PCC-4** with negative charge selectively adsorb MB

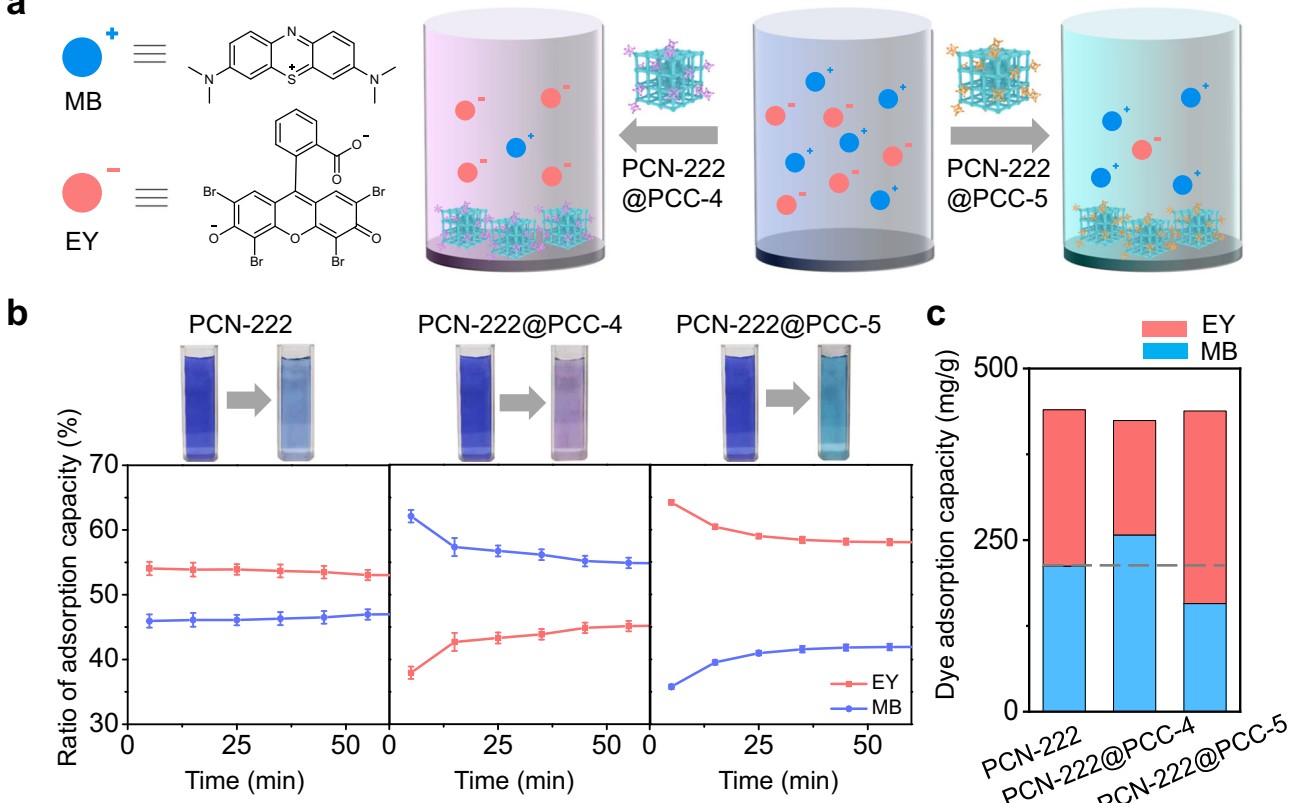

**Fig. 3 | PCN-222@PCCs mediated selective adsorption of dye molecules in aqueous solution. a** Scheme illustrating the selective dye adsorption using PCN-222@PCC-4 and PCN-222@PCC-5 in the solution containing mixed MB and EY. **b** Time-resolved dye adsorption plot of PCN-222, PCN-222@PCC-4, and PCN-222@PCC-5. Data are presented as the mean ± SD. **c** The adsorption capacity of MB and EY by PCN-222, PCN-222@PCC-4, and PCN-222@PCC-5 in the first 5 min.

dye by showing a much higher adsorption rate when compared with pristine PCN-222 and **PCN-222@PCC-5**. On the contrary, when switching the solution with negative EY, the adsorption selectivity of **PCN-222@PCCs** was fully reversed. For anionic EY, **PCN-222@PCC-5** exhibited a higher recognition rate when compared with pristine PCN-222 and **PCN-222@PCC-4** (Supplementary Figs. 26, 27). The fast kinetics can also be observed by seeing the color changing with the naked eye. FT-IR measurements were applied to characterize the **MOF@PCCs** before and after dye adsorption. Upon the introduction of dyes, the MOF@PCCs showed characteristic peaks of the dye molecules, indicating successful adsorption (Supplementary Fig. 28, dye peaks were indicated by the dashed line). In addition, the dye can be partially removed through multiple times elution, which validated the **MOF@PCCs** as reversible absorbents (Supplementary Fig. 29 and Supplementary Table 4).

Next, a mixed MB and EY dye solution were used to investigate the adsorption selectivity of pristine MOF and two MOF-PCC composites in detail. After optimization, a concentration of 12.5 mg/l of MB and 12.0 mg/l of EY solution was prepared as the stock solution, and the UV-vis adsorption spectra were recorded. When incubating 1.5 mg of PCN-222 nanocrystals in the mixed dye solution, both absorption peaks belonging to MB and EY decreased with comparable intensity (Supplementary Fig. 30). The color of the mixed dye solution also changed from deep blue to pale blue, indicating no selectivity (Fig. 3b). When replacing PCN-222 with **PCN-222@PCC-4** as the adsorbent, the color of the mixed dye solution changed to pale purple, which indicated more EY remaining in the solution (Fig. 3b). Indeed, from UV–vis adsorption spectra, the intensity of MB decreased more than that of EY, due to that **PCN-222@PCC-4** can adsorb more cationic MB, leaving anionic dye EY in the solution (Supplementary Fig. 30). In contrast, when **PCN-222@PCC-5** was introduced as the adsorbent, the color of the mixed dye solution turned pale green, indicating more MB remained (Fig. 3b). As shown in UV–vis adsorption spectra, the intensity of EY decreased higher than that of MB, which suggested that **PCN-222@PCC-5** preferably encapsulated anionic EY rather than cationic MB. The time-dependent (0–120 min) adsorption capacity curve also proves that the dye recognition selectivity of **PCN-222@PCCs** comes from the surface-coordinated PCCs (Supplementary Fig. 30). **PCN-222@PCC-4** showed a 63:37 (MB:EY) selectivity within the first 2 min, while **PCN-222@PCC-5** exhibited a reversed 35: 65 (MB:EY) selectivity (Fig. 3b). In contrast, PCN-222 showed no selectivity towards the two dye molecules with a flat curve throughout the whole period, because of the lack of change-selective interactions. Importantly, it is worth mentioning that the total dye adsorption capacity of the PCC-functionalized MOFs remained intact compared with pristine MOF, again, demonstrating that the porous cage-based capping agents can alter the microenvironment surrounding MOFs without affecting the accessible pores inside the structure (Fig. 3c and Supplementary Table 5).

## MOF@PCCs as selective catalysts in sequential reaction

Having demonstrated that MOF@PCCs possess superior selectivity against different dye molecules, next, we sought to explore the use of MOF@PCCs as selective catalysts. A typical sequential reaction, involving the hydrolysis of benzaldehyde dimethyl acetal (**1**), followed by Knoevenagel condensation with malononitrile (**2**), was chosen as the model reaction (Fig. 4a)[55–59]. By using pristine PCN-222 as the catalyst, no product selectivity was observed, generating intermediate benzaldehyde (**3**) and product benzylidenemalononitrile (**4**) in an equal amount (Fig. 4b. Interestingly, after PCC surface modification, **PCN-222@PCC-4** favored the formation of product **4** (75.4% yield and selectivity) while **PCN-222@PCC-5** selectively catalyzed the formation of intermediate **3** (89.7% yield and selectivity) (Fig. 4b, Supplementary Table 6). In contrast, other control groups including **PCN-222@PCCs** with no coordination interactions and physically mixed cages and

MOFs were recorded with minimum or no product selectivity, similar to what we observed for pristine PCN-222 (Supplementary Table 6 and Fig. 4b). The integration of PCCs and MOFs considerably enhanced the reactivity and dramatically altered the selectivity by taking the advantage of the properties of both porous materials, verifying the effectiveness of the "Cage-on-MOF" approach. The formed **MOF@PCCs** composites are also highly recyclable. The STEM images indicated no morphology change, PXRD pattern indicated the well preserved crystallinity and the ICP-OES data showed only a 15.2% cage loss after 10 runs (Supplementary Figs. 31–33 and Supplementary Table 7). The catalytic reactivity of the composites was then recorded in each run (Fig. 4c). The catalytic reactivity of the composites only declined by 15.1% after 10 cycles, which is consistent with the 15.2% loss of surface cages. These results strongly suggest that the **MOF@PCCs** retained their structural integrity during the cycles and retained most of their catalytic reactivity.

Furthermore, to determine whether PCC cavities played a crucial role in selective catalysis, we investigated the catalytic reactions when the cavities of PCCs were blocked by inhibitors. For anionic **PCC-4**, tetraphenylphosphonium chloride (TC) with a positive charge was selected as the inhibitor for the cavity (Fig. 4d). For cationic **PCC-5**, 1-adamantane carboxylic acid (AC) with a negative charge was selected as the inhibitor for the cavity (Fig. 4d). Two distinct methods were utilized to introduce the inhibitor into the cavity of the cage. The first involved soaking the PCC crystals in a solution of inhibitors, to pre-occupy the cavity of PCCs. The second involved directly adding the corresponding inhibitor to the reaction solution during the catalytic process, to compete with the potential intermediate-PCC binding. As expected, in both cases, the selectivity of **MOF@PCCs** experienced a significant decrease (Fig. 4e). When the cavities of the **PCC-4** or **PCC-5** were pre-occupied, the selectivity of **PCN-222@PCC-4** was reduced by 27.8%, and the selectivity of **PCN-222@PCC-5** was reduced by 19.3% (Fig. 4e). As a control, the inhibitors were also introduced when using the unmodified MOF (PCN-222) alone as the catalyst. The product selectivity of the inhibitor-mixed MOF remained consistent with that of the pristine MOF, suggesting that the inhibitors did not interfere with the MOF but only served to block the surface-capped PCCs (Fig. 4e).

According to time-dependent NMR monitoring and DFT calculations, a proposed reaction mechanism was developed based on the experimental and calculation results (Supplementary Figs. 34, 35 and Fig. 4f, g). Time-dependent NMR and GC analysis revealed that the MOF, **PCC-4**, and **PCC-5** promoted different steps of the reaction, leading to distinct product selectivity (Supplementary Fig. 34). The possible reaction mechanism can be proposed as follows. Initially, the α-H of the malononitrile transfers to $-SO_3^-$ of **PCC-4**, forming a malononitrile carbanion and protonated $-SO_3H$ (Supplementary Fig. 35); Subsequently, the adsorbed benzaldehyde attracts the proton of $-SO_3H$. The formed malononitrile carbanion then nucleophilically attacks the protonated benzaldehyde with accompanying dehydration, generating 2-benzylidenemalononitrile (Supplementary Fig. 35). Finally, the product is desorbed from the catalyst surface[57]. The Knoevenagel condensation reaction catalyzed by **PCC-5** undergoes a similar process (Supplementary Fig. 35). The carbanion is formed through proton transfer from malononitrile to $-NH_2$ of the **PCC-5** (Supplementary Fig. 35)[58]. The protonated benzaldehyde can also be generated by proton transfer (Supplementary Fig. 35). In both of these catalytic pathways, the critical step is the adsorption and activation of protonated benzaldehyde associated with the catalysts. In DFT calculations, it was observed that the anionic **PCC-4** exhibited high adsorption energy to protonated benzaldehyde intermediate, displaying an $E_{ads}$ of −3.7 eV, −4.3 eV, and −3.3 eV in three binding modes (Fig. 4f). The thermodynamic favorability of the adsorption process in **PCC-4** accounts for the excellent activity of **PCC-4**. However, when **PCC-4** was replaced with a cationic **PCC-5**, the positive adsorption energy ($E_{ads}$ = 3.0 eV and 2.9 EV) and long inter-molecular distance

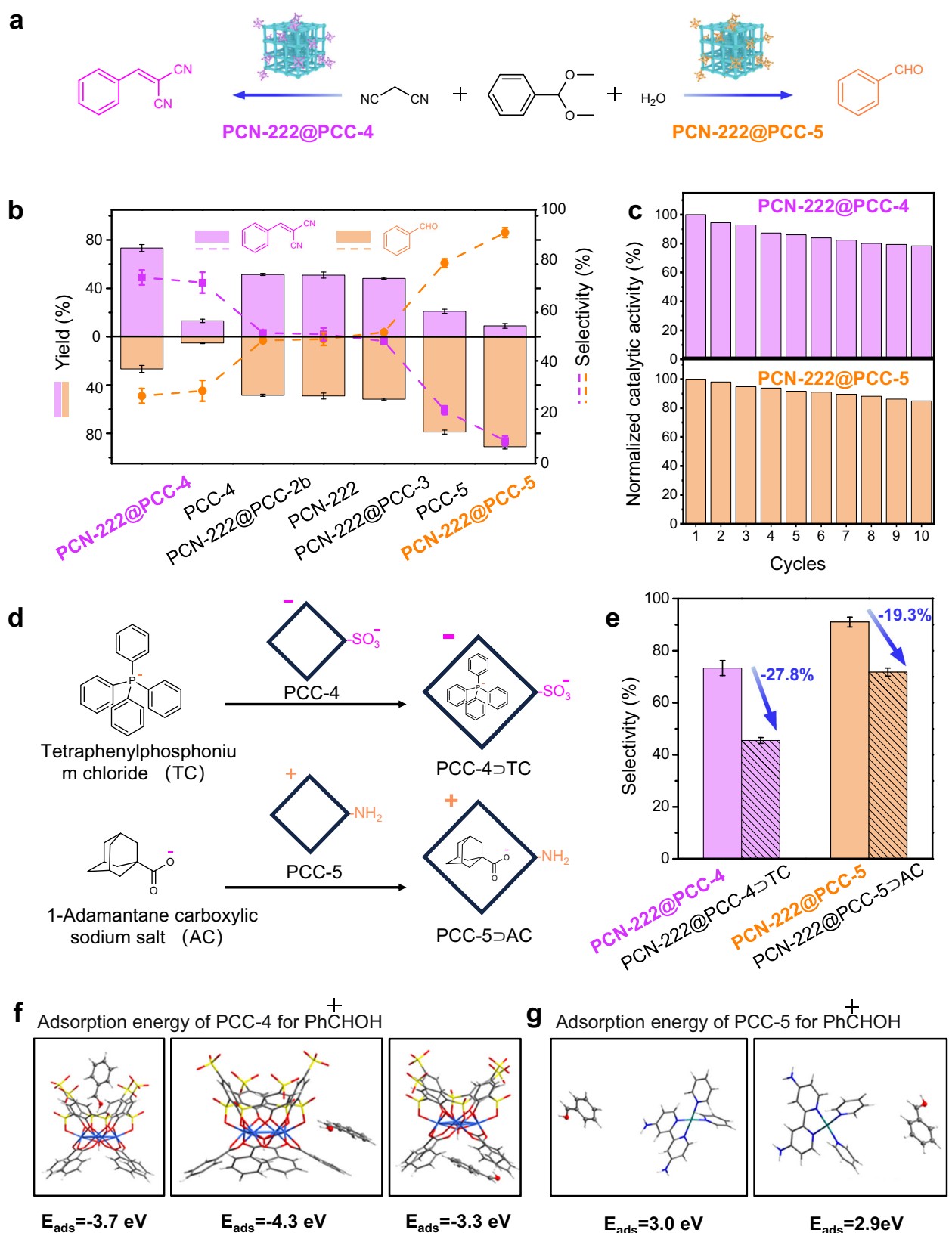

**Fig. 4 | PCN-222@PCCs mediated selective catalysis. a** Scheme illustrating the product-selective catalytic reaction. **b** Plot showing the yield and selectivity of the catalytic condensation reaction by PCN-222@PCCs and control groups. Data are presented as the mean ± SD. **c** The recyclability of PCN-222@PCCs. **d** The cavities of PCC-4 and PCC-5 were preoccupied with inhibitors. **e** Product selectivity of PCN-222@inhibited-PCCs. Data are presented as the mean ± SD. DFT calculations of the interaction modes and adsorption energies of (**f**) PCC-4 with PhCHOH and (**g**) PCC-5 with PhCHOH.

revealed very weak adsorption to the protonated benzaldehyde, resulting in no further reaction (Fig. 4g). Therefore, the sharp contrast in the binding affinity to the intermediate is believed to facilitate a distinct product selectivity of **PCC-4** and **PCC-5** modified MOF in the Knoevenagel condensation between malononitrile and benzaldehyde.

More importantly, the surface coordination of PCCs onto PCN-222 is reversible, and thereby the catalytic properties of MOF particles as heterogeneous catalysts can then be reversibly tuned by simply introducing different types of PCCs (Fig. 5a). Because the $SO_3$-Zr and $NH_2$-Zr coordination interactions between the surface-exposed metal clusters on MOF and the secondary binding sites on PCCs are weaker than the coordination bonds within the MOF, acid treatment at ambient temperature can easily remove the surface-coordinated PCCs without damaging the MOF structure. Such treatment did not change the morhology and crystallinity of the host MOF, enabling reversible modification and repeatable usage of the MOF catalysts (Fig. 5b and Supplementary Figs. 36, 37). Following the acid etching treatment, significant removal of 53.8% to 67.7% of the PCCs from the **PCN-222@PCC** system was observed, thereby indicating the surface attachment of the PCCs onto the MOF (Supplementary Table 8 and Supplementary Fig. 38). As shown in Fig. 5c, the dominant product of the sequential reaction can be reversibly tuned by modifying the same host framework with PCCs of different charges in multiple times. The product selectivity correlated well with the surface charge of the MOF nanoparticle, validating that the change in the catalytic properties comes from PCC modification (Fig. 5d). The catalytic performance of

one porous material can be reversibly tuned by surface modification of another porous material suggesting a promising route for the design of excellent heterogeneous catalysts.

## The generality of the "Cage-on-MOF" Strategy

To demonstrate the generality of the "Cage-on-MOF" strategy, we further applied it to another iconic mesoporous MOF, MIL-101. Following the abovementioned procedure, MIL-101 nanoparticles were modified with **PCC-4** and **PCC-5**, respectively. Several characterizations were performed to verify the successful formation of **MIL-101@PCCs** (Fig. 6a, c, Supplementary Figs. 39, 40). EDX element mapping also confirmed the uniform distribution of Co (II) and Pd (II) across the entire MIL-101 particle for **MIL-101@PCC-4** and **MIL-101@PCC-5**, respectively (Fig. 6b, d). The linear scanning profiles also suggested the uniform distribution of **PCC-4** and **PCC-5** throughout the entire MIL-101 particle, similar to what we observed previously in **PCN-222@PCCs** (Fig. 6e). After cage modification, **MIL-101@PCCs** adopted the same PXRD pattern and high crystallinity as the pristine MIL-101 (Fig. 6f). **PCC-4** reversed the surface charge of MIL-101 from 40 mV to -22 mV (**MIL-101@PCC-4**), while **PCC-5** further increased the positive charge to 51 mV (**MIL-101@PCC-5**) (Fig. 6g). In the XPS spectra, the downshifted binding energy of Cr (III) species after PCC coordination was observed and could be ascribed to the increase of electron density of the Cr (III) metal center after binding to the secondary coordination sites on **PCC-4** and **PCC-5** (Fig. 6h, Supplementary Fig. 41). Furthermore, the FT-IR spectra illustrated the distinctive

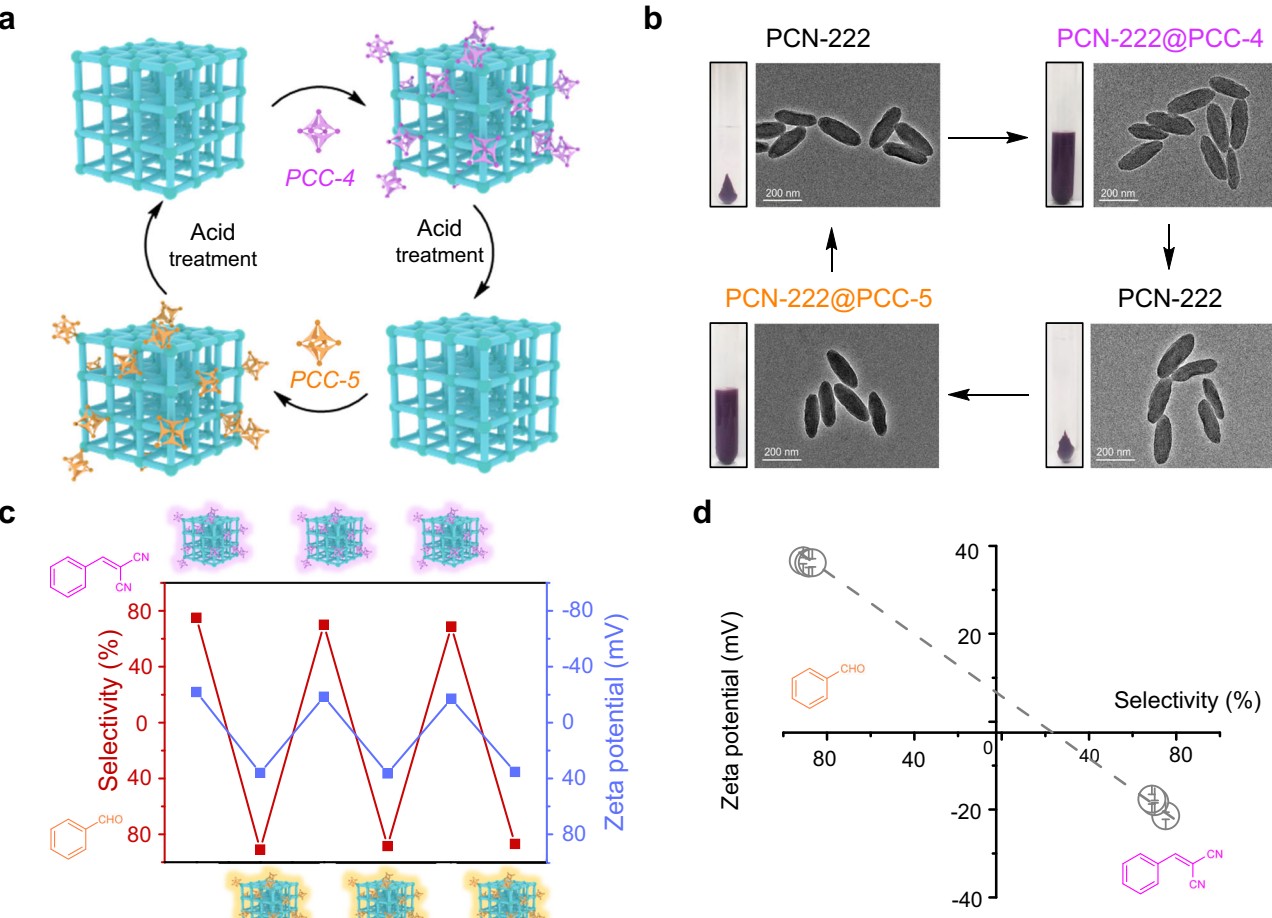

**Fig. 5 | PCN-222@PCCs as heterogeneous catalysts with reversible surface modifications. a** A schematic representation of the reversible modification of PCN-222 with PCC-4 and PCC-5. **b** TEM images and corresponding photographs showing the reversible modification process of PCN-222@PCCs. **c** The product selectivity and zeta potential of one batch of PCN-222@PCCs with recyclable surface modifications. **d** Correlation between the product selectivity and corresponding surface potential of PCN-222@PCCs. Data are presented as the mean ± SD.

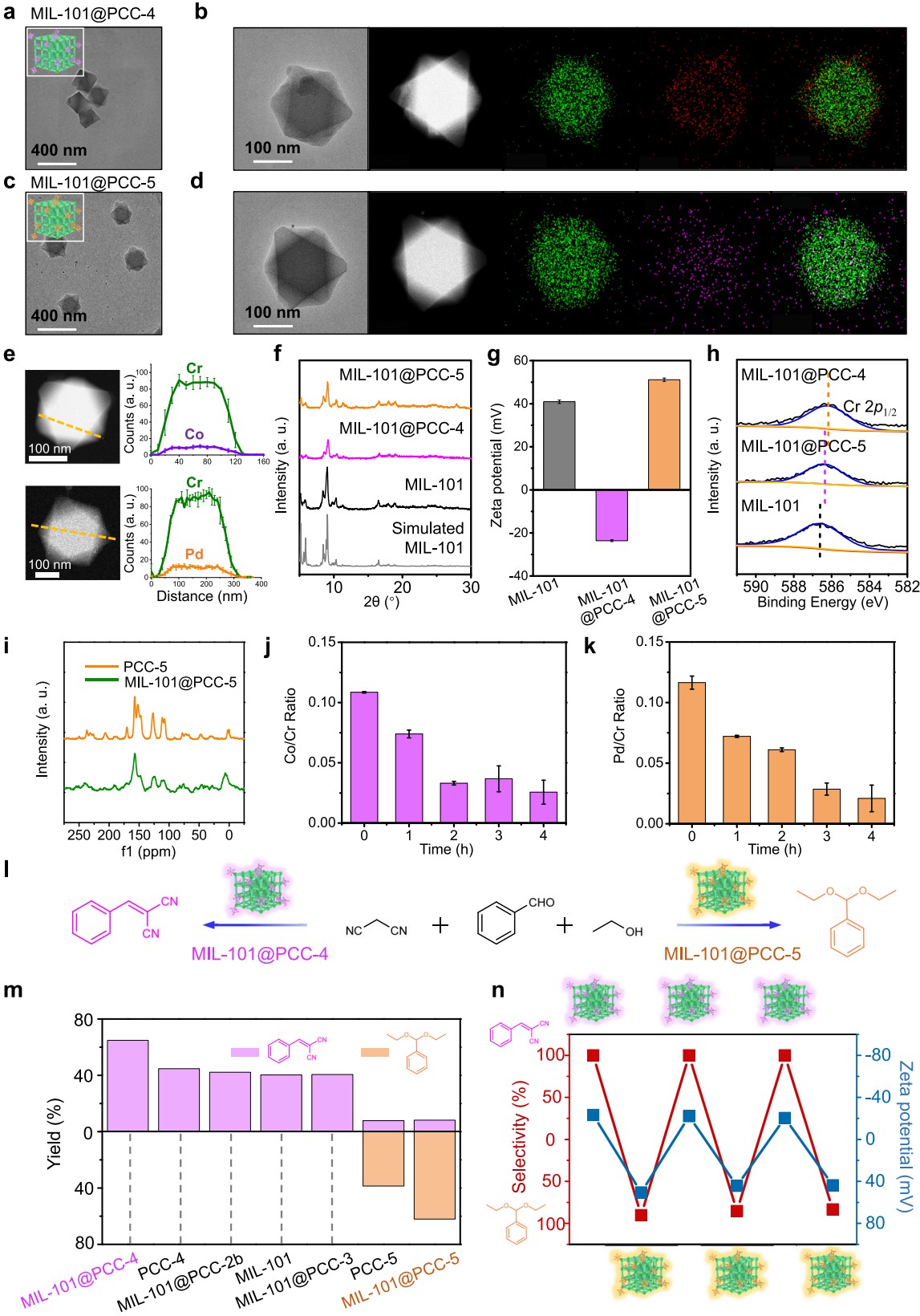

vibrational adsorption modes of the PCCs following their binding to MIL-101 (Supplementary Fig. 42). Again, **MIL-101@PCCs** exhibited preserved porosity (Supplementary Fig. 43) and enhanced chemical stability when compared with individual components in the composites (Supplementary Figs. 44–47)[58]. Quantitative ICP-OES measurements revealed the loading amount of **PCC-4** in **MIL-101@PCC-4** was

23.5 wt%, and **PCC-5** in **MIL-101@PCC-5** was 8.2 wt% (Supplementary Table 9). Theoretical calculations illustrated that PCCs coordinated with 13.3% (PCC-4) and 9.1% (PCC-5) of the surface-exposed $Cr_3O$ clusters respectively, suggesting that most PCCs located on the surface of MIL-101 (see Supporting Information Section 8 for details). However, in the case of **PCC-2b** and **PCC-3** with no secondary coordination

**Fig. 6 | Characterizations and Properties of MIL-101@PCCs.** TEM image of (**a**) MIL-101@PCC-4 and (**c**) MIL-101@PCC-5. STEM and EDX images of (**b**) MIL-101@PCC-4 and (**d**) MIL-101@PCC-5. **e** STEM linear scanning analysis of MIL-101@PCC-4 (top) and MIL-101@PCC-5 (bottom). Data are presented as the mean ± SD. **f** PXRD patterns of pristine MIL-101, MIL-101@PCC-4, and MIL-101@PCC-5. **g** Zeta potential of MIL-101, MIL-101@PCC-4 and MIL-101@PCC-5. Data are presented as the mean ± SD. **h** The *XPS* spectra show Cr $2p_{1/2}$ of MIL-101, MIL-101@PCC-4, and MIL-101@PCC-5. **i** Solid state $^{13}C$ NMR of PCC-5 and MIL-101@PCC-5. **j** Quantitative analysis of the Co/Cr ratios of MIL-101@PCC-4 under ball-milling over time. Data are presented as the mean ± SD. **k** Quantitative analysis of the Pd/Cr ratios of MIL-101@PCC-5 under ball-milling over time. Data are presented as the mean ± SD. **l** Schematic diagram of selective catalysis of a sequential condensation reaction using MIL-101@PCCs. **m** Plot showing the yield of the catalytic condensation reaction by MIL-101@PCCs and other control groups. **n** The reversible modification and recyclable selective catalysis of MIL-101@PCCs.

sites, SEM, zeta potential, PXRD, $N_2$ adsorption, and ICP-OES demonstrated that the interactions between control cages and MIL-101 were non-covalent and not stable, and thus resulted in the relatively low cage loading amounts (Supplementary Figs. 48–52 and Supplementary Table 9). The solid-state $^{13}C$ NMR spectra of PCC@MIL-101 revealed that the cage structure remained intact, resembling that of the free cage, which was consistent with previous findings in the literature (Fig. 6i)[16].

In order to further investigate the positioning of coordinated PCCs within the host MOF, we conducted ball-milling on the **MIL-101@PCCs** particles to physically remove the exterior layers of the particle. For the ball-milling experiment, the **MIL-101@PCCs** were combined with 3 mm steel balls in a planetary ball mill apparatus. Subsequently, the **MIL-101@PCCs** samples were milled for 1, 2, 3, and 4 h, followed by dispersion in ethanol for STEM characterizations. During the initial stage (0 h), the morphology of **MIL-101@PCCs** particles exhibited an octahedral shape with sharp corners. With the progression of ball-milling time, the sharp corners of **MIL-101@PCCs** particles were gradually diminished, resulting in a morphological transformation towards a rounded shape, indicating the occurrence of surface wear and tearing (Supplementary Figs. 53, 54). The results obtained from STEM elemental mapping and linear scanning analysis exhibited a reduction in the intensity of cage metal over the course of ball-milling time in both **MIL-101@PCC-4** and **MIL-101@PCC-5** samples (Supplementary Fig. 55). Moreover, the quantification of the Co/Cr and Pd/Cr ratios, which served as indicators of the cage-to-MOF ratio, exhibited a gradual decrease throughout the ball-milling process. The Co/Cr ratio decreased from approximate 0.11 to 0.026 (Fig. 6j), while the Pd/Cr ratio decreased from approximate 0.12 to 0.021 (Fig. 6k). These findings suggest that the PCCs are primarily located on the exterior surface of MOF, with fewer instances of PCC encapsulation within the MOF particles (Supplementary Fig. 55). Furthermore, we subjected **MIL-101@PCC** colloidal solutions to probe sonication at 550 W for 1 h, resulting in the formation of fractured **MIL-101@PCC** particles. Upon STEM examination, we observed significantly lower cage-to-MOF ratios in the fractured **MIL-101@PCC** compared to that in the complete particle samples, indicating that the majority of cages were attached to the external surface of the MOF (Supplementary Fig. 56). Interestingly, we observed comparable Co/Cr ratios in the complete **MIL-101@PCC** particles, regardless of whether probe sonication was applied or not, suggesting a relatively stable coordination bonding between the cages and the MOF. These findings indicate that most PCCs could remain bonded to the MOF during the probe sonication treatment (Supplementary Fig. 57).

Moreover, similar to what we observed in the case of PCN-222, **MIL-101@PCCs** particles showed significantly enhanced selectivity in dye adsorption (Supplementary Figs. 58–66 and Supplementary Tables 10, 11) as well as in the catalysis of sequential condensation reaction (Fig. 6l–n and Supplementary Table 12). **MIL-101@PCC-4** favored the formation of phenylmethylene malonitrile (64.9% yield, 100% selectivity), while **MIL-101@PCC-5** selectively catalyzed the formation of benzaldehyde diacetal (62.1% yield, 88% selectivity) (Fig. 6m and Supplementary Fig. 67)[57,59]. In addition, heterogeneous **MIL-101@PCCs** catalysts can be reused multiple times without affecting the activity, selectivity, morphology, and crystallinity (Supplementary Figs. 68–71 and Supplementary Table 13). More importantly, the

product preference can be reversibly tuned by modifying MIL-101 with different PCCs using mild acid treatment at ambient temperature to remove the attached cages (Supplementary Figs. 72–75 and Supplementary Table 14). This treatment approach, as demonstrated for **PCN-222** in Fig. 5 of the manuscript, was applied to remove coordinated **PCC-4** from **MIL-101@PCC-4** and subsequently reintroduced PCC-5 while preserving the integrity of the MOF structure. The validity of the removal and reintroduction process was further confirmed through STEM imaging, element mapping, and linear scanning analysis. The analyses conducted demonstrated a notable decrease in the Co signal after the acid treatment, followed by an observable increase in the Pd signal upon the reintroduction of **PCC-5**. These observations further support the surface attachment of PCCs onto the MOF (Supplementary Figs. 76, 77), which align well with the findings from molecular dynamics simulations (Supplementary Fig. 78). The product selectivity matched well with the surface charge of the **MIL-101@PCC** composites (Fig. 6n), demonstrating the "Cage-on-MOF" strategy is general and can be applied to different porous materials to tune the corresponding properties in a preserved porosity manner.

## Conclusions

In summary, a "Cage-on-MOF" approach was presented here to coordinatively modify MOFs with porous molecular cages to enhance their adsorption and catalytic selectivity, while retaining their accessible voids. This was achieved by functionalizing two distinct types of PCCs onto the external surface of the MOF particles through coordination interactions between the secondary coordination moieties (-$SO_3$ or -$NH_2$) on PCCs and the surface-exposed metal sites on MOFs. The resulting **MOF@PCCs** exhibited improved chemical stability, tunable surface charge, and selective dye recognition behavior, all while preserving the adsorption capacity of the parent framework. More importantly, the **MOF@PCCs** can serve as efficient heterogeneous catalysts that can achieve notable product selectivity in sequential reactions with the ability to reversibly tune the catalytic properties by altering the surface-bound cages. This study presents the first example of using a porous molecular cage as a surface capping agent for covalent functionalization of porous frameworks with switchable surface charges and controllable product selectivity. These findings pave the road to facilitate the development of advanced porous materials with enhanced properties by simply integrating existing molecular cages from the database onto diverse porous hosts.

## Methods

### Synthesis of PCC-4

Cobalt chloride hexahydrate (23.7 mg, 0.1 mmol), 2,4, 6-tri (4-carboxyl phenyl)-1,3, 5-triazine (14.5 mg, 0.33 mmol) and TSSC (18 mg, 0.015 mmol) were suspended in 2 mL MeOH. The mixture was heated up at 85 °C in an oven for 12 h. After cooling down to ambient temperature, purple crystals were collected and washed with methanol. The yield of PCC-4 (based on TSSC) was ~72.2% (according to ligand TSSC).

### Synthesis of PCC-5

To a refluxing solution of $PdCl_2$ (35.5 mg, 0.20 mmol) in MeCN (15.0 mL) was added 4,4'-diamino-2,2'-bipyridine (37.6 mg, 0.20 mmol) which was synthesized as previously reported. The resulting yellowish solution was stirred at 75 °C for another 4 h. Then the solution was

cooled down to room temperature and treated with AgPF$_6$ (55 mg, 0.40 mmol). A large amount of white precipitate appeared instantly, and the suspended solution was stirred at room temperature overnight. The white precipitate was filtrated, and the clear orange solution was evaporated to dryness to get Pd(4,4′-diamino-2,2′-bipyridine)(PF$_6$)$_2$.

Pd(4,4′-diamino-2,2′-bipyridine)(PF$_6$)$_2$ (85.2 mg, 0.30 mmol) in CH$_3$CN 15.0 ml was added 2,4,6-Tris(2-pyridyl)-s-triazine (63 mg, 0.20 mmol). The resulting yellowish solution was stirred at 60 °C for 1 h. The supernatant was recovered by centrifugation and distilled under reduced pressure to obtain PCC-5

## Synthesis of MOF@PCC-4
PCC-4 was dissolved in DMF and prepared into 1 mg/ml PCC-4 solution. And then 30 mg MOF was ultrasonically dispersed in 10 ml PCC solution, the glass reactor was sealed, and the reaction was stirred at 50 °C for 12 h. The solid product was collected by centrifugation and washed three times with DMF to remove free PCC. The resulting solid was air-dried at 60 °C.

## Synthesis of MOF@PCC-5
PCC-5 was dissolved in acetonitrile and prepared into 1 mg/ml PCC-5 solution. And then 30 mg MOF was ultrasonically dispersed in 10 ml PCC solution, the glass reactor was sealed, and the reaction was stirred at 50 °C for 12 h. The solid product was collected by centrifugation and washed three times with acetonitrile to remove free PCC. The resulting solid was air-dried at 60 °C.

## Synthesis of MOF@PCC-2b
PCC-2b was dissolved in DMF and prepared into 1 mg/ml PCC-2b solution. And then 30 mg MOF was ultrasonically dispersed in 10 ml PCC solution, the glass reactor was sealed, and the reaction was stirred at 50 °C for 12 h. The solid product was collected by centrifugation and washed three times with DMF to remove free PCC. The resulting solid was air-dried at 60 °C.

## Synthesis of MOF@PCC-3
PCC-3 was dissolved in H$_2$O and prepared into 1 mg/ml PCC-3 solution. And then 30 mg MOF was ultrasonically dispersed in 10 ml PCC solution, the glass reactor was sealed, and the reaction was stirred at 50 °C for 12 h. The solid product was collected by centrifugation and washed three times with H$_2$O to remove free PCC. The resulting solid was air-dried at 60 °C.

## X-ray single-crystal structure analyses
All crystals were taken from the mother liquid without further treatment, transferred to oil, and mounted into a loop for single-crystal X-ray data collection. Diffraction was measured on a Bruker Smart Apex diffractometer equipped with a Mo-$K_\alpha$ sealed-tube X-ray source ($\lambda = 0.71073$ Å, graphite monochromated) and a low-temperature device (110 K). The data frames were recorded using the program APEX2 and processed using the program *SAINT* routine within APEX2. The data were corrected for absorption and beam corrections based on the multi-scan technique as implemented in *SADABS*. The structures were solved by the direct method using *SHELXS* and refined by full-matrix least-squares on $F^2$ using *SHELXL* software.

## Dye adsorption of PCN-222 and PCN-222@PCCs
Eosin Y (EY) with concentrations of 10, 20, 30, 40, and 50 mg/L and methylene blue (MB) with concentrations of 10, 20, 30, 40, and 50 mg/L were prepared, respectively. Then, 2 mg of adsorbent was soaked in 20 ml of different concentrations of dye solution and stirred at room temperature for 2 h. Then the supernatant was separated by centrifugation and analyzed by UV–vis spectrum.

EY (30 mg/L) and methylene blue (10 mg/L) were respectively prepared. 2 mg of adsorbent was soaked in 20 ml dye solution, and stirred at room temperature. After a while, the supernatant was tested by UV to compare the adsorption rates of different materials

Prepare a mixed aqueous solution with both EY and MB concentrations of 12.5 mg/L, and then 1.5 mg of adsorbent was added to 40 ml of the prepared solution and stirred at room temperature. The supernatant of the solution was subjected to UV detection at regular intervals.

## Dye adsorption of MIL-101 and MIL-101@PCCs
Rhodamine B with concentrations of 20, 40, 60, 80, and 100 mg/L and methyl orange with concentrations of 20, 40, 60, 80, and 100 mg/L were prepared, respectively. Then, 5 mg of adsorbent was soaked in 20 ml of different concentrations of dye solution and stirred at room temperature for 2 h. Then the supernatant was separated by centrifugation and analyzed by UV–vis spectrum.

Rhodamine B (20 mg/L) and methyl orange (60 mg/L) were respectively prepared. 5 mg of adsorbent was soaked in 20 ml dye solution and stirred at room temperature. The supernatant of the solution was detected by UV at specific times.

Prepare a mixed aqueous solution with both Rhodamine B (120 mg/L) and methyl orange (60 mg/L), and then 5 mg of adsorbent was added to 20 ml of the prepared solution, and the supernatant of the solution was subjected to UV detection at regular intervals.

## Repeatable modification
The MOF@PCC was placed in 1 M aqueous hydrochloric acid solution and soaked for 2 h with stirring. The solids were collected by centrifugation, and the supernatant was repeatedly cleaned with deionized water until neutral. The obtained solid was immersed in DMF and stirred for 12 h. The solid was collected by centrifugation and cleaned twice with DMF to obtain MOF. The obtained MOF could be used to modify PCC again according to the above method.

## Selective catalysis of PCN-222 and PCN-222@PCCs
Benzaldehyde dimethylacetal (0.10 ml, 1.33 mmol) and malononitrile (0.13 ml, 2 mmol) were dissolved in 10 ml acetonitrile and water mixture solution ($V_{CH3CN}:V_{H2O} = 17:3$), and 10 mg catalyst was added. The reaction was heated to 50 °C and stirred for 5 h. At the end of the reaction, the supernatant was detected by gas chromatography.

## Selective catalysis of MIL-101 and MIL-101@PCCs
Benzaldehyde (0.10 ml, 1 mmol) and malononitrile (0.07 ml, 1.1 mmol) were mixed in 20 ml ethanol and DMSO solution ($V_{EtOH}:V_{DMSO} = 4:1$), 20 mg catalyst was added, and the reaction was stirred at 25 °C for 2 h. After the reaction, the supernatant was detected by gas chromatography.

## Ball-milling treatment of MIL-101@PCCs
The ball-milling process of MIL-101@PCCs involved subjecting the samples to high-energy ball milling. Specifically, 30 mg of MIL-101@PCCs and a 1 g steel ball with a diameter of 3 mm were placed within a 50 ml stainless steel pot. The milling was carried out using a planetary ball mill apparatus (BM6pro) at a rotation speed of 200 rpm and at room temperature. To prevent overheating, the milling parameters were set to operate for 2 min followed by a 4-min idle period. The total duration of active milling time was recorded as the milling time. After milling for 1 h, 2 h, 3 h, and 4 h, samples were extracted from the milling pot. These extracted samples were subsequently dispersed in ethanol using ultrasonication for the purpose of conducting STEM and EDX tests.

## Acid treatment and reversible modification

MIL-101@PCC-4 particles were immersed in a 1 M aqueous hydrochloric acid solution and stirred for 2 h. The resulting solids were separated via centrifugation, and the supernatant was repeatedly rinsed with deionized water until neutral pH was achieved. Subsequently, the obtained solids were soaked in $N,N$-dimethylformamide (DMF) for 12 h, followed by centrifugation and two washes with DMF. A portion of the solids was dispersed in ethanol for STEM and EDX analysis. STEM and EDX images revealed no significant changes in the material morphology, but the presence of Co was almost eliminated, confirming the removal of PCC-4. Next, 50 mg of MIL-101@PCC-4 after acid treatment was dispersed in a 15 ml solution of PCC-5 in acetonitrile and subjected to a reaction at 50 °C for 24 h to obtain MIL-101@PCC-5. The resulting solids were collected via centrifugation and subjected to three washes with acetonitrile. STEM and EDX analysis of a portion of the solids indicated the uniform distribution of Pd elements on the MIL-101@PCC-5 particles, suggesting the successful modification with PCC-5.

## Data availability

The data for Figs. 2–6 generated in this study are provided in the Supplementary Information/Source Data file. All data are available from the corresponding author upon request. Crystallographic data for the reported crystal structures have been deposited at the Cambridge Crystallographic Data Centre via www.ccdc.cam.ac.uk with code 2216458 (PCC-4). Correspondence and requests for materials should be addressed to Y.F. (yu.fang@hnu.edu.cn). Source data are provided with this paper.

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

## Acknowledgements

This work was financially supported by NSFC (21501133, 21877032), the China Hunan Provincial Science & Technology Department (2020RC3020, 2021JJ20021, 2022SK2003, 2022JJ10007), and the Science & Technology Innovation Program of Hunan Province (2022RC3046). We would like to thank Dr. Heng Wang from Shenzhen University for the Mass Spectrometry analysis, Prof. Hao Dong from Nanjing University for the support in molecular dynamics simulations, and Futao Huang and Yihong Zhang from Nanjing University for the measurements of linear scanning profiles using STEM. We also acknowledge Beijing Beishide Corporation for their assistance with gas adsorption analyses.

## Author contributions

Original idea was conceived by Y.F. and H.X.; synthesis was performed by Y.L. and X.Y.; X-ray crystal structure characterization was performed by Y.F. and Z.-J.G.; molecular simulations were performed by X.W.; microscope measurement was performed by Y.L. and X.Y.; dye selective adsorption was consulted by Y.L. and Z.-J.G.; catalytic reactions were conducted by Y.L.; data simulations were conduction by Z.-J.G.; data analysis was performed by Y.L.,X.Y., Y.F. and H.X.; post-synthetic modificaitons were performed by Y.F. and Y.L.; the manuscript was drafted by Y.L., H.X. and Y.F.; Y.F. and H.X. supervised the project. All authors have approved the manuscript.

## Competing interests

The authors declare no competing interests.
