## [Peer Review File · Nature Communications]

REVIEWER COMMENTS

Reviewer #1 (Remarks to the Author):

The authors report synthesis, characterization and guest-adsorption and catalytic properties of metal-organic frameworks (MOFs), PCN-222 and MIL-101 covered with porous coordination cages (PCC), PCC-4 and -5. The authors show a new strategy named "cage-on-MOF" where the adsorption capacity and catalytic reactivity of the host MOF (PCN-222) can be manipulated by exchanging the surface-mounted PCC. Although this concept is intrinsically interesting and could potentially be published in Nat. Commun., there are the following negative concerns that the authors need to address with clear experimental evidence.

First, there is insufficient experimental and theoretical supports for the changes in dye adsorption capacity and catalytic reaction in the present MOFs covered with PCCs. Figures S24 and S48 in the SI are just simple schematics that lack experimental support. Since this is at the heart of the "cage-on-MOF" strategy proposed by the authors, the authors must add the spectroscopic observations (e.g., IR absorption spectra) and crystal structure data on the guest adsorbed states, as well as data on the adsorption sites using theoretical calculations on the guest adsorbed structures. The mechanism by which the properties change with the type of PCC on the MOF surface should be discussed in details.

Second, I think the recyclability test for catalytic activity of present composites (only 3 or 5 cycles) as shown in Figures 2f and S49 is not enough because the catalytic activity continuously decreases with increasing the cycles. These results mean the mounted PCCs are gradually removed (escaped) from the surface of the MOFs. At least the authors should check the recyclability data for several ten cycles of reaction. Also, the authors should show the STEM, EDX mapping and PXRD data after recyclability tests to check the stability.

In the abstract section, the authors refer to their system as "regulators", but I think this is a bit exaggerated and inaccurate because a regulator, like a gas cylinder in general, refers to something that can control the flow rate to some degree. In the present composite, the adsorption properties and catalytic reactions certainly change depending on the type of PCC mounted on the surface, but they are not freely controllable. In this respect, I recommend that the authors change the term "regulators".

Reviewer #2 (Remarks to the Author):

Reviewer: Comments to Author:

Recommendation: The authors reported a strategy of "cage-on-MOF" to achieve selective dyes adsorption behavior and heterogeneous catalysis. They synthesized the PCC@MOFs through the binding sites, which to get a better stability than many other encapsulated "cage-MOF" materials. Moreover, they used it as catalyst to realize selective Knoevenagel condensation.

PCC, as known as metal-organic polyhedral (MOP), has received a lot of attention in recent years. Previous studies focused on the encapsulation of PCC in the cavity of host structure to form composite materials. In this manuscript, the author studied the coordination and binding of PCC on the MOF surface, and constructed an interesting surface chemistry study, which realized the unique dye adsorption performance and high selective catalytic performance. I think this research is worth to be published in Nature Communications after major revision.

1. In the nitrogen adsorption isotherm (Figure 1h), why the gas adsorption characteristics of PCN-222 with PCC surface modified are not significantly affected, and whether the modification of PCC can make the MOF surface more compact.

2. The chemical stability section, the authors soaked the PCC@MOFs in organic solvents for 24 h to prove the composite materials' stability. However, stability at higher temperatures (eg. 60 °C, 80 °C, 100 °C) is necessary, the authors should provide it.

3. The authors mentioned "PCCs without secondary coordination sites have smaller sizes and can possibly fill in the pores and block the accessible voids of the mesoporous MOFs", If the PCCs without secondary coordination sites fill in the pores of MOFs, why does nitrogen uptake of PCC-2b or PCC-3 modified MOF only decrease by 36%?

4. From the PCC loading of ICP-OES data, in the comparison of PCN-222@PCC-2b/PCC-3 and PCN-222@PCC-4/PCC-5, the content of Zr is significantly different. Why is the difference in the content of cZr so large in the same volume? Based on the synthesis section, the reaction feeding amount

of all MOF@PCCs is equivalent.

5. The positive charged PCN-222@PCC-5 has the same charge repulsion with MB, there is repulsive force between the same charges. How do dye molecules adsorb on the surface of the same charge?

6. The recyclability of materials has become a hot research topic today, whether the PCN-222@PCCs' adsorption of dye is reversible? The author should add the desorption measurement of dyes to confirm it.

7. Scale bars of SEM and TEM in the SI should be added into the images directly.

8. Please provide more about discussions on possible errors. The error bars should be added in Figure 2e and 2f.

9. The author should propose a possible illustrated reaction catalytic mechanism. It helps to understand why MOF@PCCs can achieve selective catalysis.

10. The EDX mapping of MI-101@PCCs should be provided.

11. The authors selected both PCN-222 and MIL-101 as MOFs with positive charged zeta potential. Does this strategy work equally well for MOFs with neutral and negative charged zeta potential?

12. PCN-222@PCC-4 showed an increased maximum adsorption capacity of 175 mg/g, while PCN-222@PCC-4 exhibited a decreased maximum adsorption capacity of 125 mg/g.?

Reviewer #3 (Remarks to the Author):

MOF porosity and surface decoration are widely investigated for various applications. This work demonstrates a cage-on-MOF strategy incorporating PCC on MOF for the fabrication of MOF@PCC hybrid. Specifically, PCC is introduced through coordination with metal clusters of MOF. The obtained MOF@PCC shows well-preserved or even decorated hierarchical porosity, and interactions between MOF and PCC endows it tunable electronic properties. As a result, all of the four MOF@PCC hybrids exhibit attractive selectivity in dye recognition and heterogeneous catalysis. The Introduction part attempts to signify the importance of "capping agents" in both porosity and surface decoration, but the poor readability with inappropriate description and examples make it very difficult to figure out the object and novelty of this work. As for experiments, MOF@PCC hybrids are commonly prepared through liquid-phase coordination and normally characterized. The author claimed PCC is on the exterior surface of MOF, but it is almost impossible to identify the existing state of PCC referring to the given characterizations. Further, discussions on interactions between MOF and PCC are too simple, the authors tend to ascribe the high selectivity in dye recognition and heterogeneous catalysis to the combination of MOF and PCC, whereas lacking of enough direct evidences.

This work is not qualified for publication in Nature Communications.

1. The existing state of PCC is unclear. The authors claimed PCC is coordinating to the metal clusters of MOF, in this concern PCC may exist outside MOF pores or adsorbed on MOF surfaces. Besides, according to the preparation strategy (liquid-phase coordination), it is uncontrollable hence very difficult to introduce PCC into MOF pores.

2. In Introduction, the authors tried to emphasize the significance of capping agents for selectivity promotion of the MOF composites. In this concern, capping agents encapsulated inside MOF and encapsulating MOF are both exemplified, which one is better?

3. In Introduction, the definition of capping agent is unclear. Initially, this concept is exemplified by macrocycles and polymers covering outside MOFs. Later, PCC is also included in this concept to "modify mesoporous MOFs". And this work reports the introduction of PCC via coordination. It is very difficult for readers to figure out the relationships between them.

4. In this work the cages are "demonstrated to be on the exterior surface of MOF". While Scheme 1a is signifying the introduction of Cage into MOF channels through coordination for the promotion of selectivity and stability. It is very difficult for readers to understand.

5. In Introduction, "large pore sizes generally result in poor selectivity against different substrate molecules", this description would be inappropriate.

6. TEM images at low magnification and elemental mappings could not provide direct evidence for the introduction of PCC.

7. According to N₂ adsorption isotherms, the obtained MOF@PCC composites are in microporous regime, not "hierarchical".

8. The authors claimed PCC could tune the surface charge while not providing direct evidences.

RESPONSE TO REVIEWERS' COMMENTS

Reviewer #1

The authors report synthesis, characterization and guest-adsorption and catalytic properties of metal-organic frameworks (MOFs), PCN-222 and MIL-101 covered with porous coordination cages (PCC), PCC-4 and -5. The authors show a new strategy named “cage-on-MOF” where the adsorption capacity and catalytic reactivity of the host MOF (PCN-222) can be manipulated by exchanging the surface-mounted PCC. Although this concept is intrinsically interesting and could potentially be published in Nat. Commun., there are the following negative concerns that the authors need to address with clear experimental evidence.

Response: We gratefully thank the reviewer for recognizing our work and providing constructive comments. All the comments have been addressed point-by-point in the following pages. The corresponding changes have been made in the manuscript as well as the SI.

(1) First, there is insufficient experimental and theoretical supports for the changes in dye adsorption capacity and catalytic reaction in the present MOFs covered with PCCs. Figures S24 and S48 in the SI are just simple schematics that lack experimental support. Since this is at the heart of the "cage-on-MOF" strategy proposed by the authors, the authors must add the spectroscopic observations (e.g., IR absorption spectra) and crystal structure data on the guest adsorbed states, as well as data on the adsorption sites using theoretical calculations on the guest adsorbed structures. The mechanism by which the properties change with the type of PCC on the MOF surface should be discussed in details.

Response: Thank you for the suggestions. In the revised manuscript, we have included experimental and simulation data to provide comprehensive evidence for the dye adsorption and catalysis.

Firstly, according to your suggestion, we applied FT-IR to characterize the PCC@MOF after the dye adsorption. The results were shown in Figure S26. Figure S26a showed the pristine PCN-222, PCN-222@PCC-4, and PCN-222@PCC-5 before dye adsorption. After introducing the EY and MB dyes, the characteristic peaks belonging to the dye molecules can be found in Figure S26b and Figure S26c (indicated by the black dashed line). Similarly, Figure S26d showed the pristine MIL-

101, MIL-101@PCC-4, and MIL-101@PCC-5 before dye adsorption. After introducing the MO and RhB dyes, the characteristic peaks belonging to the dye molecules can be found in **Figure S26e** and **Figure S26f** (indicated by the black dashed line). By combining FT-IR results with the UV-Vis measurements (**Figure S22-S25**), we can confirm the successful adsorption of different dye molecules within MOF@PCCs.

Figure S26. The FT-IR spectra of MOF@PCC before and after the dye adsorption. The IR spectra of (a) PCN-222, PCN-222@PCC-4, and PCN-222@PCC-5. (b) After EY adsorption. (c) After MB adsorption. (d) MIL-101, MIL-101@PCC-4, and MIL-101@PCC-5. (e) After MO adsorption. (f) After RhB adsorption.

Secondly, we applied time-dependent NMR and GC to monitor the catalytic reaction process, accompanied by the DFT calculations to investigate intermediate binding sites and to elucidate the reaction mechanism. The sequential proceeds in two steps: (1) hydrolysis of benzaldehyde dimethyl acetal to form benzaldehyde, and (2) Knoevenagel condensation of benzaldehyde and malononitrile to yield the product benzylidenemalononitrile. Given the fact that PCN-222 shows no product selectivity by forming benzaldehyde and benzylidenemalononitrile in an equal yield, the selectivity of PCN-222@PCCs originates from the different PCCs on the surface. Thus, we investigated the time-dependent reaction profiles of two steps of the reaction separately. We employed different reaction conditions of no catalyst (blank), PCN-222, PCC-4, or PCC-5 as the catalyst. In the first

step of the sequential reaction (Figure S32a), the conversion of the substrate to the product can be monitored by the decrease of “proton a” and increase of “proton b” (Figure S32b-e). From the time-dependent conversion plot (Figure S32f), it is clear that without a catalyst (blank), the reaction can proceed 25% after 5 hrs. However, by using PCC-4 as the catalyst, the conversion was reduced to 15% after 5 hrs, which proved that the anionic cage will inhibit the reaction. In contrast, both PCN-222 and PCC-5 dramatically enhanced the reactivity by showing 100% conversion within 4 hrs, indicative of the reactivity promotion. Compared with PCN-222, the cationic PCC-5 can reach the quantitative yield within 2 hrs, by showing faster kinetics than that of PCN-222. In the second step of the sequential reaction (Figure S32g), the conversion of the substrate to the product can be monitored by the decrease of “proton b” and increase of “proton c” (Figure S32h-k). From the time-dependent conversion plot (Figure S32l), it is clear that without a catalyst (blank) the reaction can only proceed 40% after 5 hrs. However, by using PCC-5 as the catalyst, the conversion was reduced to 10% after 5 hrs, which proved that the cationic cage will inhibit the reaction. PCN-222 catalyst can give a moderate yield of 55% after 5 hrs. In contrast, by using anionic PCC-4 as the catalyst, the conversion was increased to 80% after 3 hrs, indicating the enhancement of the reactivity. By taking a close look at the reaction profile (Figure S32f & S32l), it is demonstrated that the anionic PCC-4 inhibits the first step but promotes the second step, whereas cationic PCC-5 enhances the first step but prohibits the second step. Thus, by employing different catalysts in the separated steps of the sequential reaction, PCC was found to be the origin of the product selectivity.

Figure S32. Time-dependent reaction profiles of the sequential catalytic reaction monitored by NMR and GC. (a) Reaction scheme of the first step of the sequential reaction. NMR of the reaction profile after 1h, 3 h, and 5 h in the presence of blank (b), PCN-222 (c), PCC-4 (d), and PCC-5 (e). (f) Substrate conversion plot in the presence of the different catalysts. (g) Reaction scheme of the second step of the sequential reaction. NMR of the reaction profile after 1h, 3 h, and 5 h in the presence of blank (h), PCN-222 (i), PCC-4 (j), and PCC-5 (k). Substrate conversion plot in the presence of a different catalyst (l).

Then, we proposed the reaction mechanism based on the experimental results, previously reported studies, and DFT calculations. In MOF@PCC4-mediated catalysis (Figure S33), the substrates (malononitrile and benzaldehyde) was absorbed on the surface of the catalyst. Initially, the α -H of the malononitrile is transferred to $-\text{SO}_3^-$ of the cage to form a malononitrile carbanion and protonated $-\text{SO}_3\text{H}$; subsequently, the proton of $-\text{SO}_3\text{H}$ was abstracted by the adsorbed benzaldehyde. Then, the formed malononitrile carbanion nucleophilic attack protonated benzaldehyde with accompanying dehydration to produce 2-benzylidenemalononitrile. Finally, the

product is desorbed from the catalyst surface. The Knoevenagel condensation reaction catalyzed by MOF@PCC-5 undergoes a similar process (Figure S33), where the carbanion is formed through proton transfer from malononitrile to -NH_2 of the catalyst, and the protonated benzaldehyde is also generated by proton transfer.

(a) PCN-222

(b) PCN-222@PCC-4

(c) PCN-222@PCC-5

Figure S33. Diagram of the proposed catalytic reaction mechanism of PCN-222 (a), PCN-

222@PCC-4 (b), PCN-222@PCC-5 (c).

In these two catalytic pathways, the critical step is the adsorption and activation of protonated benzaldehyde on the catalysts. The DFT results indicate that PCC-4 has significantly higher adsorption energy for protonated benzaldehyde than PCC-5 (Figure 3j & 3k). The thermodynamic favorability of the adsorption process in PCC-4 is responsible for the excellent activity of PCC-4. In sharp contrast, PCC-5 gives positive adsorption energy, indicating an extremely weak adsorption ability to protonated benzaldehyde, and thereby resulting in its poor catalytic activity in the Knoevenagel condensation between malononitrile and benzaldehyde.

Figure 3. The DFT calculated adsorption energies of the intermediates in the presence of PCC-4 (j) and PCC-5 (k).

Furthermore, in order to investigate the role of the cavity of the PCC in selective catalysis, we investigated the catalytic reactions by using PCCs with their cavities blocked by inhibitors. For anionic PCC-4, positive tetraphenylphosphonium chloride (TC) was selected as the inhibitor for the cavity (Figure 3d). For cationic PCC-5, negative 1-adamantane carboxylic acid (AC) was selected as the inhibitor for the cavity (Figure 3d). We employed two different methods to introduce the inhibitor into the cavity of the cage: (1) soaking the PCC crystals in the solution of the inhibitors to pre-occupy the cavity of PCCs, and (2) during the catalytic process, directly adding the corresponding inhibitor to the reaction solution to compete with the PCC. As expected, in both of the cases, the selectivity of MOF@PCC was significantly decreased (Figure 3e). Pre-occupying the cavities of PCC-4 or PCC-5 reduced the selectivity of PCN-222@PCC-4 from 75.4% to 44.5%, the selectivity of PCN-222@PCC-5 was reduced from 89.7% to 71.8% (Figure 3e). As a control, the inhibitors were added when employing the MOF (PCN-222) alone as the catalyst, and the product selectivity remained as the MOF catalyst alone (52.7%), indicating that there was no interference from the inhibitor (Figure 3b).

Figure 3. (a) Scheme illustrating the product-selective catalytic reaction. (b) Plot showing the yield and selectivity of the catalytic condensation reaction by using PCN-222@PCCs and control groups. (c) The recyclability of PCN-222@PCCs. (d) Pre-occupying the cavities of PCC-4 or PCC-5 using inhibitors. (e) Product selectivity of PCN-222@PCCs with inhibitors and PCN-222@PCCs.

Finally, in our attempts to obtain crystal structures of MOF@PCCs with encapsulated dye molecules or substrates, we encountered challenges due to the random attachment of PCCs to the MOF surface through coordination with exposed metal clusters. As a result, the long-range order of MOF@PCCs was diminished, making it impossible to determine the single crystal of MOF@PCC, let alone those with guest molecules. In addition, the high symmetry of the PCN-222 and MIL-101 (cubic space group) presents difficulties in determining of low symmetrical guests (triclinic space group), such as the dye molecules or substrates within the pores of the MOF.

In summary, based on the results discussed above, we applied the experimental data (FT-IR, NMR, GC, and control experiments) and simulation results to fully support our claims about dye adsorption and selective catalysis in the manuscript.

(2) Second, I think the recyclability test for catalytic activity of present composites (only 3 or 5 cycles) as shown in Figures 2f and S49 is not enough because the catalytic activity continuously decreases with increasing the cycles. These results mean the mounted PCCs are gradually removed (escaped) from the surface of the MOFs. At least the authors should check the recyclability data for several ten cycles of reaction. Also, the authors should show the STEM, EDX mapping, and PXRD data after recyclability tests to check the stability.

Response: Thank you for the suggestions. In the revised manuscript, we extended the recycling process to ten cycles and evaluated the catalytic reactivity (Figure 2c and Figure S62) using PXRD (Figure S30 and S64), STEM (Figure S31 and S65) and ICP-OES (Table S7 and Table S13).

We performed ten successive recycling runs of PCN-222@PCC-4, PCN-222@PCC-5, MIL-101@PCC-4, and MIL-101@PCC-5 and investigated the catalytic reactivities for each run (Figure 2c and Figure S62). The results revealed that even after five cycles, the catalytic reactivity remained at 91.6% of the pristine catalyst. After ten cycles, the catalytic reactivity decreased slightly to 84.9% of the pristine catalyst. These data strongly support that the MOF@PCC remained the structural integrity the recycling process and most of the catalytic reactivity was preserved.

Figure 2c. The catalytic activity of PCN-222@PCC-4 and PCN-222@PCC-5 after 10 times of cycles.

Figure S62. The catalytic activity of MIL-101@PCC-4 and MIL-101@PCC-5 after 10 times of cycles.

PXRD verified that all four catalysts retained a high crystalline state after ten cycles when compared with the pristine catalyst (Figure S30 and S64).

Figure S30. PXRD of PCN-222@PCC-4 and PCN-222@PCC-5 after 10 times of cycles.

Figure S64. The PXRD of MIL-101@PCC-4 and MIL-101@PCC-5 after 10 times of cycles.

STEM and EDX analysis also proved that the morphology of the MOF@PCC remained the same as the pristine catalyst even after ten cycles (Figure S31 and S65). There is no breakage or surface damage to the particle.

Figure S31. The STEM images of PCN-222@PCCs after 10 cycles.

Figure S65. The STEM images and EDX of MIL-101@PCCs after 10 cycles.

ICP-OES analysis was applied to quantitatively study the elemental change after recycling. Table R1 demonstrated that the weight loss of surface PCC is in the range of 8.9%-17.8%. The maximum loss of 17.8% of the surface PCC (Table S7 and Table S13) is consistent with the catalytic reactivity loss of about 20% (Table S7 and Table S13). This will-matched data again proved the product selectivity is governed by the surface PCCs.

Table S7. ICP-OES data of PCN-222@PCCs after 10 cycles.

	V ₀ (mL)	C _{Zr} (mg/L)	C _{Co} (mg/L)	C _{Pd} (mg/L)	PCC/PCN- 222@PCC (wt%)	The amount of PCC loss (wt%)
PCN-222@PCC-4	10	10.20	0.29	/	5.12	8.9
PCN-222@PCC-5	10	10.02	/	0.10	1.68	15.2

Table S13. ICP-OES data of MIL-101@PCCs after 10 cycles.

	V ₀ (mL)	C _{Cr} (mg/L)	C _{Co} (mg/L)	C _{Pd} (mg/L)	PCC/MIL- 101@PCC (wt%)	The amount of PCC loss (wt%)
MIL-101@PCC-4	10	9.42	1.28	/	16.73	17.8
MIL-101@PCC-5	10	10.01	/	0.47	7.21	12.3

(3) In the abstract section, the authors refer to their system as “regulators”, but I think this is a bit exaggerated and inaccurate because a regulator, like a gas cylinder in general, refers to something that can control the flow rate to some degree. In the present composite, the adsorption properties and catalytic reactions certainly change depending on the type of PCC mounted on the surface, but they are not freely controllable. In this respect, I recommend that the authors change the term "regulators".

Response: Thank you for the clarification. We have changed the term "regulator" to "modulator" in the revised manuscript in response to your comments. Considering the term "modulator" has been widely used in nanoscale MOF synthesis to describe agents used to alter the size and surface properties of nanoparticles, we believe that "modulator" is an appropriate term to describe the effect that PCCs mounted on MOF surfaces have on the catalytic and adsorption properties of the host framework. The term “regulator” has been changed to “modulator” in the revised manuscript as well as in the revised SI.

Reviewer #2:

Reviewer: Comments to Author:

Recommendation: The authors reported a strategy of “cage-on-MOF” to achieve selective dyes adsorption behavior and heterogeneous catalysis. They synthesized the PCC@MOFs through the binding sites, which to get a better stability than many other encapsulated “cage-MOF” materials. Moreover, they used it as catalyst to realize selective Knoevenagel condensation.

PCC, as known as metal-organic polyhedral (MOP), has received a lot of attention in recent years. Previous studies focused on the encapsulation of PCC in the cavity of host structure to form composite materials. In this manuscript, the author studied the coordination and binding of PCC on the MOF surface, and constructed an interesting surface chemistry study, which realized the unique dye adsorption performance and high selective catalytic performance. I think this research is worth to be published in Nature Communications after major revision.

Response: We gratefully thank the reviewer for recognizing our work and providing constructive comments. All the comments have been addressed point-by-point in the following pages. The corresponding changes have been made in the manuscript as well as the SI.

(1) In the nitrogen adsorption isotherm (Figure 1h), why the gas adsorption characteristics of PCN-222 with PCC surface modified are not significantly affected, and whether the modification of PCC can make the MOF surface more compact.

Response: Thank you for the comments. We attributed the preserved gas adsorption properties of PCN-222 with PCC surface modification to the maintained porosity of the PCN-222@PCCs composites. To demonstrate this is the case, we performed the following experiments, and the results were briefly summarized below.

We used quantitative ICP-OES measurements together with the theoretical calculation based on the crystal structures to illustrate the possible arrangement modes of PCCs on the surface of MOF particles. According to ICP-OES results, the mass percentage of PCC mounted on the exterior surface of PCN-222@PCC is less than 6% (5.4% for PCC-4 and 1.98% for PCC-5), indicating surface coating of PCC rather than filling into the pores (Table S3). By using the crystal structure data and the mass percentage results of the PCC, the theoretical calculation revealed that there is on average approximately 1 PCC molecule per 10 surface Zr_6 clusters with a surface density of ca.

10.50% (PCC-4) and 8.83% (PCC-5). The results suggest the majority of the MOF particle surface is exposed with the open void channels intact, thus leading to the preserved gas adsorption characteristics of PCN-222@PCCs (Figure 1i).

We further used MD simulation to illustrate the PCC coverage on the MOF surface to provide direct visualization of the surface coating mode. As can be seen in Figure 11-m, both PCC-4 and PCC-5 molecules solely attach to the surface but not inside the pores of PCN-222 due to the steric hindrance and electrostatic repulsion. Additionally, because of the relatively low surface density and the porosity of the molecular cage, the open void channels of MOF are not blocked, thus not affecting the adsorption of gas molecules.

Figure 1. Characterization of the PCN-222@PCCs. TEM images of PCN-222@PCC-4 (a) and PCN-222@PCC-5 (c). STEM image and EDX element mapping of a single particle of PCN-222@PCC-4 (b) and PCN-222@PCC-5 (d). PXRD patterns (e), Zeta potential measurements (f), and FT-IR spectra (g) of pristine PCN-222, PCN-222@PCC-4, and PCN-222@PCC-5. The XPS spectra show Zr 3d_{3/2} and 3d_{5/2} binding energy of PCN-222, PCN-222@PCC-4, and PCN-222 (h). N₂ adsorption isotherms with N₂ uptake normalized to the same mass of PCN-222 (i). The surface density of PCN-222@PCCs (j). Photostability of PCN-222@PCC in benzylamine methanol solution (k). Molecular dynamics simulations of PCN-222@PCC-4 (l) and PCN-222@PCC-5 (m).

The above-mentioned experiments and theoretical calculations have been included in the main text and in SI. To better clarify this point, we have added/revised the sentences in the main text as follows:

“The number of modified cages on the MOF surface was quantified by inductively coupled plasma optical emission spectrometry (ICP-OES). The weight percentage of the two PCCs in PCN-222@PCC-4 and PCN-222@PCC-5 were measured to be 5.40 wt% and 1.98 wt%, respectively (Table S3). By calculating the surface density, the surface PCC per Zr₆ cluster in PCN-222@PCC-4 and PCN-222@PCC-5 was identified as 10.50% and 8.83%, respectively (Figure 1j). After the surface modification of PCCs, the chemical and thermal stability of prepared MOF@PCC composites was greatly enhanced (Figure 1k and Figure S13-S15). In addition, the binding interaction between the surface PCCs and internal MOFs is strong enough for further applications, such as reversible dye adsorption and heterogeneous catalysis (Figure S16)”

“As further support for PCC surface binding on MOF, MD simulations revealed that PCC-4 and PCC-5 were unable to enter the pores of PCN-222 due to steric hindrances and electrostatic repulsions (Figure 1l and 1m). The surface binding mode of PCCs was further confirmed by MD simulations, which showed almost no cages entered pores in a ca. 10 nm-sized lattice system (see Supporting Information Section 4)”

(2) The chemical stability section, the authors soaked the PCC@MOFs in organic solvents for 24 h to prove the composite materials' stability. However, stability at higher temperatures (eg. 60 °C, 80 °C, 100 °C) is necessary, the authors should provide it.

Response: Thank you for the suggestions. In this revision, we investigated the chemical stability of PCCs (Figure S14) and MOF@PCCs (Figure S15) at different temperatures.

Firstly, the stability of PCC alone was studied at 25°C, 40°C, 60°C, 80°C, and 100°C. It was found that PCC-4 remained stable under 25-80°C, while gradually decomposing at 100°C by showing a distinct UV-Vis spectrum (Figure S14a). In contrast, PCC-5 remained stable under 25-100°C by showing a consistent UV-vis peak (Figure S14b).

Figure S14. The stability of PCC-4 and PCC-5 at different temperatures.

Then, the stability of PCN-222@PCCs and MIL-101@PCCs was studied at 25°C, 40°C, 60°C, 80°C, and 100°C by soaking the materials in the solvent and characterizing the supernatant by UV-Vis. For PCN-222@PCC-4 under 25-80°C, no characteristic peak belonging to PCC-4 was found, indicating the absence of PCC leaching or decomposing (Figure S15a). When the temperature was elevated to 100°C, little amount of decomposed PCC-4 was found in the supernatant (Figure S15a), which well-matched the results discussed above (Figure S15a). For MIL-101@PCC-5 under 25-60°C, no characteristic peak belonging to PCC-4 was found, indicating no PCC leaching or decomposing (Figure S43a). When the temperature was elevated to 80°C, little amount of intact PCC-4 was found in the supernatant. If the temperature was further elevated to 100°C, a characteristic peak belonging to decomposed PCC-4 was found (Figure S43a). MOF@PCC-5 exhibited different behaviors. In the temperature range of 25-100°C, no characteristic peak belonging to PCC-5 was found, suggesting no PCC-5 leaching or decomposing (Figure S15b & S43b).

Figure S15. The stability of PCN-222@PCCs at different temperatures.

Figure S43. The stability of MIL-101@PCCs at different temperatures.

The results suggest that MOF@PCCs can maintain their structural integrity at temperatures ranging from 25 to 80°C, while gradual decomposition occurs at 100°C. It is noteworthy that the catalytic reactions were conducted below 50° C, where the MOF@PCC catalysts remained intact.

(3) The authors mentioned “PCCs without secondary coordination sites have smaller sizes and can possibly fill in the pores and block the accessible voids of the mesoporous MOFs”, If the PCCs without secondary coordination sites fill in the pores of MOFs, why does nitrogen uptake of PCC-2b or PCC-3 modified MOF only decrease by 36%?

Response: Thank you for the comments. Because of the small sizes of PCC-2b and PCC-3 without secondary coordination sites, both molecular cages can enter the MOF channel. However, since the interactions between PCC-2b or PCC-3 to the parent MOF framework are relatively low due to the

lack of interactions, only about 1 wt% of the cages are recorded to be functionalized in the framework determined by ICP-OES measurements, which is only about 1 cage molecule per 421 unit cells. The low loading of PCC-2b and PCC-3 can only slightly decrease the void space and surface area of parent MOF but cannot completely block all the accessible voids.

The ICP-OES results and the calculation process are shown in Figure S17-S20 and Table S3 in the SI. We have also included following texts to briefly explain the reason.

“In contrast, without secondary coordinative groups, PCC-2b and PCC-3 can only be encapsulated within the pore of the MOFs (Figure S17-S20). As a result, the N₂ uptake capacity was significantly decreased by ca. 36% after PCC-2b or PCC-3 modification on PCN-222 (Figure S20), indicative of a diminishing of the accessible voids. Given the lack of sufficient binding interactions, the loading amount on MOF was determined to be only ca. 10.7 mg/g for PCC-2b and 8.2 mg/g for PCC-3, which were considerably lower than that of their congeners with secondary coordination sites (Table S3 and Figure S21).”

(4) From the PCC loading of ICP-OES data, in the comparison of PCN-222@PCC-2b/PCC-3 and PCN-222@PCC-4/PCC-5, the content of Zr is significantly different. Why is the difference in the content of cZr so large in the same volume? Based on the synthesis section, the reaction feeding amount of all MOF@PCCs is equivalent.

Response: Thank you for the comments. The observed significant difference in the Zr content is because of the different dilution factors used in our previous experiments. While this variation does not impact the relative ratio between Zr/Co or Zr/Pd, it may present some difficulty for readers to understand. In this revision, we have conducted a fresh round of the ICP-OES measurements and have presented the results Table S3.

Table S3. ICP-OES of MOF@PCCs.

	V ₀ (mL)	C _{Zr} (mg/L)	C _{Co} (mg/L)	C _{Pd} (mg/L)	PCC/PCN- 222@PCC (wt%)
PCN-222@PCC-4	10	24.24	0.76	/	5.40
PCN-222 mixed with PCC-4	10	12.33	0.10	/	1.46

PCN-222@PCC-2b	10	11.12	0.07	/	1.02
PCN-222@PCC-5	10	21.97	/	0.26	1.98
PCN-222 mixed with PCC-5	10	13.67	/	0.13	1.60
PCN-222@PCC-3	10	12.93	/	0.08	0.79

(5) The positive charged PCN-222@PCC-5 has the same charge repulsion with MB, there is repulsive force between the same charges. How do dye molecules adsorb on the surface of the same charge?

Response: Thank you for the comments. In general, guest molecules are adsorbed inside the accessible voids of MOF mainly via Van der Waals interactions, rather than solely on the exterior surface of MOF particles. Both PCN-222 and MIL-101 have large surface areas, high porosity, and thus high surface energy, which can be reduced by adsorbing the guest substances. The use of PCN-222 and MIL-101 as positively charged MOFs for adsorbing cationic dyes has been reported in the published literature (*Chem. Eng. J.*, **2020**, 399, 125346 and *J. Mater. Chem. A*, **2015**, 3, 1675–1681). As we mentioned in previous Question #3, the mass percentage of PCC mounted on the exterior surface of PCN-222 is about 5%, which is ca. 1 PCC molecule per 10 Zr₆ clusters on the particle surface. The incorporation of PCC-5 is solely on the exterior surface of MOF particles and only slightly increases the surface charge of these two MOFs, but does not change the interior porous nature of the MOF materials. Therefore, the guest dye molecules can still get into MOF open channels and bind to the framework.

(6) The recyclability of materials has become a hot research topic today, whether the PCN-222@PCCs' adsorption of dye is reversible? The author should add the desorption measurement of dyes to confirm it.

Response: Thank you for the comments.

We performed recyclability studies in the revised manuscript. The MOF@PCCs exhibit high recyclability in catalytic reactions, as demonstrated in Figure 2c and S62. However, their recyclability for dye adsorption is moderate (Figure S27, Figure S57, Table S4, and Table S10).

To investigate this question, we conducted the experiments as follows: 2 mg PCN-222@PCCs were suspended in 40 mL cationic dye solution (Eosin Y and Methylene Blue) with a concentration of 40 mg/L. 5 mg MIL-101@PCCs were suspended in a 40 mL anionic dye solution (Rhodamine B and Methylene orange) with a concentration of 40 mg/L. After staying still for 2 hrs, the solid was collected by centrifugation. After dye adsorption, PCN-222@PCCs were washed by MeCN and the solution was concentrated for UV-Vis analysis (Figure S27). For MIL-101@PCCs with adsorbed dye, they were washed with the mixed solvent of MeCN and MeOH ($V_{\text{MeCN}} : V_{\text{EtOH}}=1:1$) and further treated with UV-Vis analysis (Figure S57). As a control, pristine PCN-222 was conducted with the same procedure for the two types of dye molecules. It was found that most cationic dyes can be efficiently desorbed, however, only part of the anionic dyes can be removed. For Methylene Blue (cationic, +1), the release percentage for PCN-222, PCN-222@PCC-4, and PCN-222@PCC-5 is 65.5%, 72.7%, and 63.0%, respectively. For Rhodamine B (cationic, +1), the release percentage for MIL-101, MIL-101@PCC-4, and MIL-101@PCC-5 is 71.4%, 81.5%, and 70.7%, respectively. In contrast, for Eosin Y (anionic, -2), the release percentage for PCN-222, PCN-222@PCC-4, and PCN-222@PCC-5 is 26.2%, 27.8%, and 34.7%, respectively (Table S4). For Methyl Orange (anionic, -1), the release percentage for MIL-101, MIL-101@PCC-4, and MIL-101@PCC-5 is 18.7%, 22.7%, and 20.2%, respectively (Table S10). For PCN-222 alone, it cannot release all of the dyes because the high surface area and porosity will interact with the adsorbed dye molecules, thus retaining part of the encapsulated molecules within the surface or the pores.

Figure S27. UV-vis spectra of dye molecules released from PCN-222@PCCs. (a) MB. (b) EY.

Figure S57. UV-vis spectra of dye molecules released from **MIL-101@PCCs**. (a) RhB. (b) MO.

Table S4. Dye release from **PCN-222@PCCs**.

Materials	Adsorption capacity of MB (mg)	The release of MB (mg)	Release rate of MB (%)
PCN-222	0.29	0.19	65.5
PCN-222@PCC-4	0.33	0.24	72.7
PCN-222@PCC-5	0.27	0.17	63.0
Materials	Adsorption capacity of EY (mg)	The release of EY (mg)	Release rate of EY (%)
PCN-222	0.64	0.17	26.5
PCN-222@PCC-4	0.54	0.15	27.8
PCN-222@PCC-5	0.75	0.27	34.7

Table S10. Dye release from **MIL-101@PCCs**.

Materials	Adsorption capacity of RhB (mg)	The release of RhB (mg)	Release rate of RhB (%)
MIL-101	0.42	0.30	71.4
MIL-101@PCC-4	0.54	0.44	81.5

MIL-101@PCC-5	0.41	0.29	70.7
Materials	Adsorption capacity of MO (mg)	The release of MO (mg)	Release rate of MO (%)
MIL-101	0.75	0.14	18.7
MIL-101@PCC-4	0.44	0.10	22.7
MIL-101@PCC-5	0.79	0.16	20.2

(7) Scale bars of SEM and TEM in the SI should be added into the images directly.

Response: Scale bars have been added to the SEM and TEM images in the revised SI.

(8) Please provide more about discussions on possible errors. The error bars should be added in Figure 2e and 2f.

Response: We have added error bars to Figure 2e and 2f. Based on multiple experimental runs, the observed variations are minimal.

(9). The author should propose a possible illustrated reaction catalytic mechanism. It helps to understand why MOF@PCCs can achieve selective catalysis.

Response: Thank you for the comments. We added the description in the main text and SI as follows:

"According to time-dependent NMR monitoring and DFT calculations, a proposed reaction mechanism was developed based on the experimental and calculation results (Figure S32 & S33 and Figure 3j & 3k). Time-dependent NMR and GC analysis revealed that the MOF, PCC-4, and PCC-5 promoted different steps of the reaction, leading to distinct product selectivity (Figure S32). The possible reaction mechanism can be proposed as follows. Initially, the α -H of the malononitrile transfers to $-\text{SO}_3^-$ of PCC-4, forming a malononitrile carbanion and protonated $-\text{SO}_3\text{H}$ (Figure S33); Subsequently, the adsorbed benzaldehyde attracts the proton of $-\text{SO}_3\text{H}$. The formed malononitrile carbanion then nucleophilically attacks the protonated benzaldehyde with accompanying dehydration, generating 2-benzylidenemalononitrile (Figure S33). Finally, the product is desorbed from the catalyst surface. ^[53] The Knoevenagel condensation reaction catalyzed by PCC-5 undergoes a similar process (Figure S33). The carbanion is formed through proton transfer from

malononitrile to -NH_2 of the PCC-5 (Figure S33).^[54] The protonated benzaldehyde can also be generated by proton transfer (Figure S33). In both of these catalytic pathways, the critical step is the adsorption and activation of protonated benzaldehyde associated with the catalysts. In DFT calculations, it was observed that the anionic PCC-4 exhibited high adsorption energy to protonated benzaldehyde intermediate, displaying an E_{ads} of -3.7 eV, -4.3 eV, and -3.3 eV in three binding modes (Figure 3j). The thermodynamic favorability of the adsorption process in PCC-4 accounts for the excellent activity of PCC-4. However, when PCC-4 was replaced with a cationic PCC-5, the positive adsorption energy ($E_{\text{ads}} = 3.0$ eV and 2.9 eV) and long inter-molecular distance revealed extremely weak adsorption to the protonated benzaldehyde, resulting in no further reaction (Figure 3k). Therefore, the sharp contrast in the binding affinity to the intermediate is believed to facilitate a distinct product selectivity of PCC-4 and PCC-5 modified MOF in the Knoevenagel condensation between malononitrile and benzaldehyde.”

(a) PCN-222

(b) PCN-222@PCC-4

(c) PCN-222@PCC-5

Figure S33. Diagram of the catalytic reaction mechanism of PCN-222 (a), PCN-222@PCC-4 (b), PCN-222@PCC-5 (c).

(10). The EDX mapping of MI-101@PCCs should be provided.

Response: Thank you for the suggestions. The TEM, STEM, and EDX mapping of MI-101@PCCs have been added in the revised main text (Figure 5).

Figure 5. Using the "Cage-on-MOF" strategy to engineer MIL-101 nanoparticles. TEM image of MIL-101@PCC-4 (a) and MIL-101@PCC-5 (c). STEM and EDX images of MIL-101@PCC-4 (b) and MIL-101@PCC-5 (d).

(11) The authors selected both PCN-222 and MIL-101 as MOFs with positive charged zeta potential. Does this strategy work equally well for MOFs with neutral and negative charged zeta potential?

Response: Thank you for the suggestions.

The interaction between the molecular cages and the parent MOF particles is established through a coordination bond between the secondary coordination sites on the molecular cage and the exterior surface-exposed open metal sites on the MOF. Thus, this strategy works equally for different types of MOFs with neutral and negative surface charges.

To demonstrate this is the case, we modified PCCs onto the negatively charged MIL-125-NH₂ using the same modification strategy. As shown in Figure R1d, after modifying with PCC-4, the surface potential of MIL-125-NH₂ was reduced from -18.1 mV to -20.3 mV. While after modifying with PCC-5, the surface potential of MIL-125-NH₂ increased to +14.6 mV.

To further demonstrate the generality of this strategy, we further applied this strategy to incorporate molecule cages onto the surface of other four distinct MOFs with different surface charges (Figure R1-2). For all tested MOFs, their zeta potentials showed significant changes after PCC modification. Further STEM characterizations revealed that PCCs were uniformly mounted onto MOF particle surfaces without changing their morphologies (Figure R2). Therefore, we believe that this “Cage-on-MOF” strategy is generally applicable to different types of MOFs no matter the surface charge due to the strong coordination interactions.

Figure R1. Zeta potential of UiO-66@PCCs, MIL-101(Fe)@PCCs, ZIF-8@PCCs, and MIL-125-NH₂@PCCs.

Figure R2. TEM image and EDX of UiO-66@PCCs, MIL-101(Fe)@PCCs 和 ZIF-8@PCCs.

(12) PCN-222@PCC-4 showed an 232 increased maximum adsorption capacity of 175 mg/g, while PCN-222@PCC-4 exhibited a decreased 233 maximum adsorption capacity of 125 mg/g.?

Response: Thank you for the comments.

Modifying different PCCs on MOFs affects only the dye selectivity, not the adsorption capacity. It was found that the total dye adsorption capacity remained almost the same for PCN-222, PCN-222@PCC-4, and PCN-222@PCC-5. This is because that less than 20 wt% of PCCs were modified on the surface of parent MOF, which had a minimal impact on the total porosity. Thus, intrinsic porosity of the MOF played the dominant role in dye adsorption, with the PCC governing which type of dye can enter. Indeed, there were slight variations in the adsorption capacity of the cationic and anionic dyes. This was due to the highly negative surface charge of PCN-222@PCC-4 (zeta potential = -20 mV), which resulted in strong interaction with the cationic dyes, resulting in a slightly higher maximum adsorption capacity for cationic dyes than for anionic dyes.

Reviewer #3:

MOF porosity and surface decoration are widely investigated for various applications. This work demonstrates a cage-on-MOF strategy incorporating PCC on MOF for the fabrication of MOF@PCC hybrid. Specifically, PCC is introduced through coordination with metal clusters of MOF. The obtained MOF@PCC shows well-preserved or even decorated hierarchical porosity, and interactions between MOF and PCC endows it tunable electronic properties. As a result, all of the four MOF@PCC hybrids exhibit attractive selectivity in dye recognition and heterogeneous catalysis.

The Introduction part attempts to signify the importance of “capping agents” in both porosity and surface decoration, but the poor readability with inappropriate description and examples make it very difficult to figure out the object and novelty of this work. As for experiments, MOF@PCC hybrids are commonly prepared through liquid-phase coordination and normally characterized. The author claimed PCC is on the exterior surface of MOF, but it is almost impossible to identify the existing state of PCC referring to the given characterizations. Further, discussions on interactions between MOF and PCC are too simple, the authors tend to ascribe the high selectivity in dye recognition and heterogeneous catalysis to the combination of MOF and PCC, whereas lacking of enough direct evidences.

Response: We gratefully thank the reviewer for recognizing our work and providing constructive comments. All the comments have been addressed point-by-point in the following pages. The corresponding changes have been made in the manuscript as well as the SI.

(1) The existing state of PCC is unclear. The authors claimed PCC is coordinating to the metal clusters of MOF, in this concern PCC may exist outside MOF pores or adsorbed on MOF surfaces. Besides, according to the preparation strategy (liquid-phase coordination), it is uncontrollable hence very difficult to introduce PCC into MOF pores.

Response: Thank you for the comments. Briefly, because of the size and surface charge of the PCC we used in this study, the PCCs with secondary coordination sites were mounted onto the exterior surface of the MOF particles. In our revised manuscript, we carried out the following experiments, including ICP-OES (Table S3), STEM (Figure 5i), solid-state NMR (Figure 5h), UV-vis (Figure S16 and S44), and MD simulation (Figure 2 and Figure S70), to study the state of PCC with the

parent MOF particle.

First, we used quantitative ICP-OES measurements together with the theoretical calculation based on the crystal structures to illustrate the possible arrangement modes of PCCs on the surface of MOF particles. According to ICP-MS results, the mass percentage of PCCs mounted on the exterior surface of PCN-222@PCC is less than 6% (5.4% for PCC-4 and 1.98% PCC-5), indicating surface coating of PCC rather than filling into the pores (Table S3). By using the crystal structure data and the mass percentage results of the PCC, the theoretical calculation revealed that there is on average approximately 1 PCC molecule per 10 Zr₆ clusters on the surface. The weight percentages of the two PCCs in PCN-222@PCC-4 and PCN-222@PCC-5 were determined to be 5.40 wt% and 1.98 wt%, respectively (Table S3). Based on the calculated surface density, the molar ratio of surface-loaded PCCs versus Zr₆ clusters in PCN-222@PCC-4 and PCN-222@PCC-5 was identified as 10.50% and 8.83%, respectively (Figure 1j). These results indicate that the majority of the PCC molecules serve as the surface capping agents to be mounted onto the exterior surface of MOF particles rather than to get inside the pores of the parent framework.

Table S3. ICP-OES of MOF@PCCs.

	V ₀ (mL)	C _{Zr} (mg/L)	C _{Co} (mg/L)	C _{Pd} (mg/L)	PCC/PCN- 222@PCC (wt%)
PCN-222@PCC-4	10	24.24	0.76	/	5.40
PCN-222 mixed with PCC-4	10	12.33	0.10	/	1.46
PCN-222@PCC-2b	10	11.12	0.07	/	1.02
PCN-222@PCC-5	10	21.97	/	0.26	1.98
PCN-222 mixed with PCC-5	10	13.67	/	0.13	1.60
PCN-222@PCC-3	10	12.93	/	0.08	0.79

Second, we used STEM to test the element mapping of MIL-101@PCCs particles before and after probe sonication to disrupture the particle (Figure 5i-j). Ideally, if the PCCs are mostly mounted on the surface of the particle, the cage-to-MOF ratio will be lower in the MIL-101@PCCs fractures

as compared to in the complete MIL-101@PCCs, since few PCCs are encapsulated inside MOF particle. To rupture the MOF particle, we treated the MIL-101@PCCs solution using a probe sonicator with an energy of 550 W for 1 h. As can be seen in Figure 5j, there is a low Co signal recorded for the fractured MIL-101@PCCs sample. The quantification of the Co/Cr intensity ratio (as an indicator of the cage-to-MOF ratio) revealed that the Co/Cr ratio of fractured MIL-101@PCCs is significantly lower than that of the complete particle samples, suggesting most Co-cage shall be attached to the external particle surface. Additionally, both complete MIL-101@PCCs samples with and without probe sonication treatment showed similar Co/Cr ratio, indicating the coordination bonding of cage onto MOF is relatively stable and most PCCs can stay with MOF under probe sonication (Figure S51).

Figure 5. Solid state ¹³C NMR of PCC-5 and MIL-101@PCC-5 (h). The EDX of MIL-101@PCCs particles before and after fracturing by probe sonicator (i). The comparison of the Co/Cr ratio on the exterior surface before and after fracturing (j).

Figure S51. The EDX element mapping of complete MIL-101@PCC-4 samples with and

without probe sonication treatment (a). Comparison of the Co/Cr ratio of MIL-101@PCC-4 samples with and without probe sonication treatment (b).

Third, we further used MD simulation to illustrate the PCC coverage on the MOF surface to provide direct visualization of the surface coating mode. As demonstrated by MD simulation, both PCC-4 and PCC-5 molecules solely attach onto the exterior surface of PCN-222 as well as MIL-101 but not inside pores, due to the steric hindrance and electrostatic repulsion (Figure 2 and Figure S70). The spacing filling mode of the MOF particle surface is the relatively low surface of the molecular cages which are mostly attaching to the surface but not getting into the open channels.

Figure 2. Molecular dynamics simulations of PCN-222@PCC-4 (l) and PCN-222@PCC-5 (m).

Figure S70. Simulation structure of MIL-101@PCC-4 and MIL-101@PCC-5 on the (111) facet.

Furthermore, we used UV-vis (Figure S16 and S44) to study the state of the MOF@PCC solution. We tried to separate the PCCs from the surface of MOF@PCC by repeatedly sonicating it and testing the supernatant solution with UV-vis measurements. As shown in Figure S16 and S44, the characteristic peak of the separated PCC did not change, indicating that the PCC remained stable in the form MOF@PCC.

Figure S16. UV detection of the state of the stripped PCCs.

Figure S44. UV detection of the state of the stripped PCCs.

Finally, regarding the reviewer's concern about the "liquid-phase coordination" method, some previously published results have used a similar synthetic route to use small-sized cages without coordination groups as the functionalization agents (*J. Am. Chem. Soc.* **2019**, 141, 12182–12186 and *Angew. Chem. Int. Ed.* **2021**, 60, 14138–1414). These cages were demonstrated to enter the

cavities but not on the exterior surface with little interaction with the parent MOF due to the small size and lack of secondary coordination sites. In this study, we highlight the use of a molecular cage with a relatively large size and secondary coordination sites as the porous surface capping agent to modify the exterior surface of parent MOFs instead of the interior space. The designed coordination groups on PCCs enable the formation of coordination bonds, thus allowing a specific and stable surface functionalization.

(2) In Introduction, the authors tried to emphasize the significance of capping agents for selectivity promotion of the MOF composites. In this concern, capping agents encapsulated inside MOF and encapsulating MOF are both exemplified, which one is better?

Response: Thank you for the comments. The use of PCCs to modify MOF particles in the pores has been demonstrated in the previous study and this work (*J. Am. Chem. Soc.* **2016**, *138*, 1138–1141 and *ACS Appl. Mater. Interfaces* **2019**, *11*, 12639–12646). In our case, we succeeded in modifying the exterior of the MOF surface through coordination interactions. Both ways have been demonstrated with the ability to improve the selectivity of parent MOF materials for adsorption and catalysis applications. However, if the PCCs are encapsulated inside MOF, the intrinsic porosity of the MOF particle will be affected, leading to a decreased surface area and adsorption capacity. However, the “Cage-on-MOF” strategy we reported in this study only incorporates porous PCCs onto the MOF surface to promote selectivity without sacrificing the MOF porosity by blocking the cavities.

(3) In Introduction, the definition of capping agent is unclear. Initially, this concept is exemplified by macrocycles and polymers covering outside MOFs. Later, PCC is also included in this concept to “modify mesoporous MOFs”. And this work reports the introduction of PCC via coordination. It is very difficult for readers to figure out the relationships between them.

Response: Thank you for the comments. To better clarify the “Cage-on-MOF” approach, in the revised manuscript, we clearly state that the PCC molecules are mounted on the exterior surface of the MOF particle via coordination with the surface exposed metal sites. According to the experimental and theoretical data we provided in Question #1, we have provided fairly solid evidence to illustrate that PCCs are mostly on the surface of “mesoporous MOFs” PCN-222 and

MIL-101, due to the relatively large size, secondary coordination sites, and surface charges of the PCCs. Thus, we choose to use the term “surface modulator” and “surface capping agent” in the revised manuscript to describe the roles of PCCs in the modification of MOF particles.

The revised sentences about PCCs are highlighted in the main text and listed below:

“To introduce additional functions onto MOFs without compromising their porosity, a surface capping agent should be carefully designed to possess intrinsic porosity that permits guest molecule transport, as well as tunable functions that enable specific recognition.”

“However, the majority of “Cage-MOF” hybrid materials reported so far have been formed through non-covalent encapsulation, which is likely to affect the porous structure of the hybrid material due to the random arrangement of cages within the host voids. Furthermore, non-covalent interactions may also pose a concern regarding the stability of the “Cage-MOF” hybrid material in extreme conditions (Scheme 1a).^[39, 40] “

(4) In this work the cages are “demonstrated to be on the exterior surface of MOF”. While Scheme 1a is signifying the introduction of Cage into MOF channels through coordination for the promotion of selectivity and stability. It is very difficult for readers to understand.

Response: Thank you for the comments. We have revised the Scheme 1a to better illustrate the location of PCCs on the exterior surface of MOF particles. The revised Scheme 1a is shown below.

Scheme 1 | Illustration of the “Cage-on-MOF” strategy. (a) Advantages of cage-coordinated MOF compared with pristine MOF, polymer-modified MOF, and cage-encapsulated MOF. (b) X-ray single crystal structure of PCC-4 and PCC-5 with secondary coordination sites. The cartoon for showing the cage-coordinated MOF composites: MOF@PCC-4 and MOF@PCC-5.

(5) In Introduction, “large pore sizes generally result in poor selectivity against different substrate molecules”, this description would be inappropriate.

Response: Thank you for the comments. We have changed the context in the main text as follows:

“However, large pore sizes are generally associated with poor selectivity against different substrate molecules, leading to undesirable side products as well as poor separation performance (Scheme 1a).^[20]”

(6). TEM images at low magnification and elemental mappings could not provide direct evidence for the introduction of PCC.

Response: Thank you for the comments. In addition to STEM images and element mapping, we

used solid-state NMR (Figure 5h) and ICP-OES measurements to quantitatively demonstrate the introduction of PCCs (Table S3).

In the case of PCN-222, we recorded and calculated from ICP-MS results that there are ca. 539 PCC-4 molecules and 454 PCC-5 molecules per 200 nm PCN-222 particle, respectively. Similarly, in the case of MIL-101, there are ca. 28,431 PCC-4 molecules and 19,706 PCC-5 molecules per 200 nm MIL-101 particle, respectively.

The introduction of PCCs has also been demonstrated by the solid-state NMR results and the semi-quantitative analysis of the ^{13}C NMR spectrum is consistent with what we obtained using ICP-OES data.

Figure 5 (h) Solid state ^{13}C NMR of **PCC-5** and **MIL-101@PCC-5**.

Table S3. ICP-OES data of **PCN-222@PCCs**

	V_0 (mL)	C_{Zr} (mg/L)	C_{Co} (mg/L)	C_{Pd} (mg/L)	PCC/PCN-222@PCC (wt%)
PCN-222@PCC-4	10	24.24	0.76	/	5.40
PCN-222 mixed with PCC-4	10	12.33	0.10	/	1.46
PCN-222@PCC-2b	10	11.12	0.07	/	1.02
PCN-222@PCC-5	10	21.97	/	0.26	1.98
PCN-222 mixed with PCC-5	10	13.67	/	0.13	1.60

PCN-222@PCC-3	10	12.93	/	0.08	0.79
---------------	----	-------	---	------	------

(7) According to N₂ adsorption isotherms, the obtained MOF@PCC composites are in microporous regime, not “hierarchical”.

Response: Thank you for the comments. You are correct that the major pores in MOF@PCCs are micropores. However, in some cases, they can be regarded as “hierarchical porous materials”, as long as they have two types of pores (micropores and mesopores). Herein, both MIL-101 and PCN-222 are defined as “hierarchical porous MOF” in the literature (*Angew. Chem. Int. Ed.* **2019**, 58, 10104–10109). In addition, PCC-4 and PCC-5 both have a mesoporous cavities. Thus, when combining the different pores from the MOF and the PCC, the MOF@PCC composite can be also regarded as a “hierarchical porous material”. Furthermore, in this paper, we aim to emphasize the concept of “using porous materials to modify porous MOFs”, thereby obtaining a composite with enhanced properties without diminishing the porosity.

(8) The authors claimed PCC could tune the surface charge while not providing direct evidences.

Response: Thank you for the comments. In the main text, we provided the zeta potential for the MOF@PCCs. It shows the surface charge was reversed significantly (Figure 1f).

Figure 1. Characterization of the PCN-222@PCCs. TEM images of PCN-222@PCC-4 (a) and PCN-222@PCC-5 (c). STEM image and EDX element mapping of a single particle of PCN-222@PCC-4 (b) and PCN-222@PCC-5 (d). PXRD patterns (e), Zeta potential measurements (f), and FT-IR spectra (g) of pristine PCN-222, PCN-222@PCC-4, and PCN-222@PCC-5. The XPS spectra show Zr 3d_{3/2} and 3d_{5/2} binding energy of PCN-222, PCN-222@PCC-4, and PCN-222 (h). N₂ adsorption isotherms with N₂ uptake normalized to the same mass of PCN-222 (i). The surface density of PCN-222@PCCs (j). Photostability of PCN-222@PCC in benzylamine methanol solution (k). Molecular dynamics simulations of PCN-222@PCC-4 (l) and PCN-222@PCC-5 (m).

Furthermore, the charge reverse is not limited to PCN-222 and MIL-101. Other MOFs, including UiO-66, UiO-67, and MIL-125-NH₂ can also be modified with cages and their charge can be tuned

accordingly (Figure R3).

Figure R3. Zetal potential of pristine MOFs and MOF@PCCs.

REVIEWER COMMENTS

Reviewer #1 (Remarks to the Author):

The topic of this paper is the synthesis, characterization and guest-adsorption and catalytic properties of metal-organic frameworks (MOFs), PCN-222 and MIL-101 covered with porous coordination cages (PCC), PCC-4 and -5. I think the authors made an effort to respond to the reviewer's comments. The requests made by the reviewers were addressed appropriately and the manuscript was much improved. I think this paper is publishable in Nat. Commun.

Reviewer #2 (Remarks to the Author):

In this revision, the authors did make some progress on the manuscript. There are still some problems to be solved before it can be published.

1. Per the authors' revision, the loading of PCCs on the surface of MOFs is just 1 wt%, why does such a low loading cause a significant selectivity and catalysis performance? The authors mentioned that "the mass percentage of PCC mounted on the exterior surface of PCN-222@PCC is less than 6%", I think this capacity is difficult to achieve such performance. More discussions should be provided.

2. Even without considering the size of the cage, how can ensure that the cage can be evenly distributed on the MOF surface? I suggest authors make a line scan spectrum (on TEM) of one PCC@MOFs to confirm the PCC just on the surface of MOFs and can evenly distribute.

Reviewer #3 (Remarks to the Author):

The authors have addressed most of my concerns. But there are still some issues need revision.

1. It is highly recommended for the authors to provide high resolution TEM images and elemental mappings to directly confirm the introduction of PCC. Besides, no SEM images could be found in Figure 1.

2. It is very difficult to understand the so-called "hierarchical" nature of PCN-222 and MOF@PCC, the given N₂-sorption isotherms imply they are in microporous regime.

3. The authors claimed they tried to emphasize the concept "using porous materials to modify porous MOFs", then it may be inappropriate or unconvincing to employ a microporous PCC to modify the so-called "hierarchical PCN-222" (even if it is hierarchical porous) to obtain "enhanced properties".

RESPONSE TO REVIEWERS' COMMENTS

Response to Comments of Reviewer #1:

General Comments:

The topic of this paper is the synthesis, characterization and guest-adsorption and catalytic properties of metal-organic frameworks (MOFs), PCN-222 and MIL-101 covered with porous coordination cages (PCC), PCC-4 and -5. I think the authors made an effort to respond to the reviewer's comments. The requests made by the reviewers were addressed appropriately and the manuscript was much improved. I think this paper is publishable in Nat. Commun.

Response:

We express our gratitude to the reviewer for acknowledging our work. We appreciate all the efforts made by the reviewer throughout the review process.

Response to Comments of Reviewer #2:

General Comments:

In this revision, the authors did make some progress on the manuscript. There are still some problems to be solved before it can be published.

Comment 1. Per the authors' revision, the loading of PCCs on the surface of MOFs is just 1 wt%, why does such a low loading cause a significant selectivity and catalysis performance? The authors mentioned that "the mass percentage of PCC mounted on the exterior surface of PCN-222@PCC is less than 6%", I think this capacity is difficult to achieve such performance. More discussions should be provided.

Response:

We appreciate the reviewer for highlighting this important aspect. It is indeed noteworthy that despite the relatively low content of approximately 5 wt% and 10 mol% of PCCs on the PCN-222 MOF surface, they exert a notable influence on the properties and applications of the MOFs. This intriguing and distinctive characteristic of surface modifications has been further elaborated upon in the main text to mitigate any potential misunderstandings.

1. One of the primary factors contributing to the relatively low loading but significant impact of PCCs is the surface binding mode between PCCs and MOF particles. The quantification of PCC loading on the MOF was determined using ICP-OES measurements and theoretical evaluation. In the Supplementary Information (Sections 4 and 8), we described the ICP-OES measurements and theoretical evaluation procedures. Based on these analyses, we proposed that PCCs exhibit an exterior surface modification mode, with approximately 5-6 wt% and 10 mol% of PCCs loading on the MOF particle. In this mode, ideally, every 10 metal clusters on the MOF

surface support a single PCC molecule. From this perspective, the loading of PCCs at 5-6 wt% is substantial, as it could effectively cover the majority of the surface area on a single MOF particle. In addition, the relatively low mass percentage can partially be attributed to the low density of PCCs.

2. Existing literature suggests that the performance of MOF nanoparticles is primarily influenced by their surface properties. Some reports indicate that the introduction of functional molecules exclusively on the surface of MOF particles, even at low weight percentages (5%-15 wt%), can induce significant changes in various physical and chemical properties, such as solubility, surface adhesion, stability, gas adsorption, and catalytic properties. (See *Angew. Chem. Int. Ed.* 2023, 62, e202303280; *J. Am. Chem. Soc.* 2022, 144, 685–689; *J. Am. Chem. Soc.* 2021, 143, 13557; *Chem. Sci.* 2019, 10, 3289–3294). An illustrative example is the work by Yaghi et al. where they demonstrated the use of amino acid-functionalized MOF for enhancing CO₂ capturing (*J. Am. Chem. Soc.* 2022, 144, 2387–2396). It was found that only **9.8 wt%** amino acid incorporation can lead to a substantial **53.9% increase** in the CO₂ adsorption capacity of MOF-808. Another noteworthy example comes from the Li group, who demonstrated the exceptional dispersibility of a polymer-coated MOF in various solvents. In this case, a surface coating of just 2.7 wt% significantly influenced the suspension behavior of the targeted MOFs. These literature reports clearly demonstrate that despite the relatively low mass loading of surface modification motifs, these molecules still exert a significant impact on the properties and functionalities of the MOFs.

3. In this study, the underlying mechanism for selective adsorption primarily relies on the alteration of surface charge through the surface modification of positively or negatively charged PCCs. The modification of the negatively charged cage (PCC-4) led to a change in the surface charge of PCN-222 from about +20.7 to -21.6, as determined by zeta potential measurements. In contrast, the functionalization with positively charged PCC-5 further increased the surface charge of PCN-222 to about +27.3. The significant contrast in surface charge (-21.6 vs. +27.3) of the MOF leads to the selective adsorption of organic dyes. Similarly, the mechanism for selective catalysis can be attributed to the selective stabilization of intermediate molecules. Consistent with previously reported observations in the literature, these performance characteristics are surface-dependent, thus indicating that the presence of PCCs coating on the MOF surface is significant in generating such enhanced performance.

In order to further elucidate the significance of PCC surface modification in influencing the properties of MOF particles, we have expanded the discussion in the main text, specifically on **Pages 8-9 in the manuscript**. This additional content provides a more comprehensive insight into the effects of PCC surface modification on the overall behavior and characteristics of the MOF particles.

Comment 2. Even without considering the size of the cage, how can ensure that the cage can be evenly distributed on the MOF surface? I suggest authors make a line scan spectrum (on TEM) of one PCC@MOFs to confirm the PCC just on the surface of MOFs and can evenly distribute.

Response:

We appreciate the valuable suggestions provided by the reviewer. Following the reviewer's comment, we conducted STEM element linear scanning on four MOF@PCC samples (**Figure R1 and Figure R2**), namely PCN-222@PCC-4, PCN-222@PCN-5, MIL-101@PCN-4, and MIL-

101@PCN-5. Additionally, we carried out ball-milling and acid treatment experiments on MIL-101@PCC samples to eliminate the PCCs immobilized on the surface. Subsequently, the resulting samples were subjected to STEM element mapping and linear scanning analysis to demonstrate that the PCCs are primarily attached to the external surface.

Figure R1. STEM images and linear scanning analysis with normalized intensities of (a) PCN-222@PCC-4, (b) PCN-222@PCC-5, (c) MIL-101@PCC-4, and (d) MIL-101@PCC-5.

Figure R2. Examinations of the (a) Zr/Co and (b) Zr/Pd element ratios on PCN-222@PCCs samples. The curves demonstrated a lower Zr/PCCs ratio at the thin edge and a higher Zr/PCC ratio at the thick center.

Figure R1 depicts the results of the STEM element linear scanning for all four MOF@PCCs particles. These results provided clear evidence of the uniform distribution of Co (from PCC-4) and Pd (from PCC-5) elements throughout the entirety of the MOF particle with noticeable lower PCC metal intensity compared to the MOF, thereby suggesting the homogeneous immobilization of PCCs onto the surface of host framework. Importantly, a further examination of the Zr/Co and Zr/Pd element ratios on PCN-222@PCCs samples, which indicate the proportion of metal clusters on the MOF relative to the cage molecules in the linear scanning profiles from the particle's edge to its center, revealed a distinct volcano-shaped curve. This curve demonstrated a lower Zr/PCC ratio at the thin edge and a higher Zr/PCC ratio at the thick center, thereby suggesting that the immobilization of PCCs likely occurs on the surface of the framework rather than being embedded within it (**Figure R2**).

Figure R3. STEM images, elemental mapping and linear scanning analysis of MIL-101@PCC-4 particles following various durations of ball-milling. (a) 0 h, (b) 1 h, (c) 2 h, (d) 3 h, and (e) 4 h.

In addition, to investigate the positioning of coordinated PCCs within the MOF, we conducted ball-milling on the MIL-101@PCC particles to physically remove the exterior layers of the particles (**Figure R3-R5**). For the ball-milling experiment, the MIL-101@PCCs were combined with 3 mm steel balls in a planetary ball mill apparatus. Subsequently, the MIL-101@PCCs samples were milled for durations of 1 h, 2 h, 3 h, and 4 h, followed by dispersion in ethanol for STEM characterizations. During the initial stage (0 h), the morphology of MIL-101@PCC-4 nanoparticles

exhibited an octahedral shape with sharp corners (Figure R3a). With the progression of ball-milling time, the sharp corners of MIL-101@PCC-4 nanoparticles were gradually removed, resulting in a transformation of their morphology into a rounded shape (Figure R3b-e), indicating surface wear and tearing. The results obtained from STEM elemental mapping and linear scanning analysis exhibited a reduction in the intensity of cage metal over the course of ball-milling in both MIL-101@PCC-4 and MIL-101@PCC-5 samples (Figure R3 and R4). A similar trend was observed for MIL-101@PCC-5 (Figure R4). Moreover, the quantification of the Co/Cr and Pd/Cr ratios, which serve as indicators of the cage-to-MOF ratio, demonstrated a gradual decline during ball-milling. The Co/Cr ratio decreased from approximately 0.11 to 0.026 (Figure R5a), while the Pd/Cr ratio decreased from around 0.12 to 0.021 (Figure R5b). Further calculations of the PCC modification percentages as compared to initial states following ball-milling revealed that less than 25% of PCCs can be recorded, indicating that the majority of PCCs are predominantly situated on the exterior surface of the MOF, with fewer PCCs encapsulation within the MOF particles (Figure R5).

Figure R4. STEM images, elemental mapping and linear scanning analysis of MIL-101@PCC-5 particles following various durations of ball-milling. (a) 0 h, (b) 1 h, (c) 2 h, (d) 3 h, and (e) 4 h.

Figure R5. Quantitative analysis of (a) the Co/Cr ratios and PCC-4 modification percentage of MIL-101@PCC-4 as well as (b) the Pd/Cr ratios and PCC-5 modification percentage of MIL-101@PCC-5 under ball-milling over time.

Figure R6. A schematic representation, STEM images, elemental mapping, and linear scanning analysis of the (a) PCN-222@PCC-4, (b) PCN-222@PCC-4 after acid treatment, and (c) reversible modification of PCC-5 on PCN-222.

Figure R7. STEM images and linear scanning analysis of the (a) PCN-222@PCC-4, (b) PCN-222@PCC-4 after acid treatment, and (c) reversible modification of PCC-5 on PCN-222.

Furthermore, we conducted an acid treatment at ambient temperature to selectively eliminate the surface-coordinated PCCs from MIL-101 while preserving the integrity of the MOF structure. This treatment approach, as demonstrated for PCN-222 in Figure 4 of the manuscript, was applied to remove PCC-4 from MIL-101@PCC-4 and subsequently reintroduce PCC-5. The validity of this removal and reintroduction process was confirmed through STEM imaging, element mapping, and linear scanning analysis. These analyses revealed a significant decrease in the Co signal following the acid treatment, and a subsequent increase in the Pd signal upon the reintroduction of PCC-5. These findings provide evidence of the surface attachment of PCCs onto the MOF (Figure R6 & R7).

As a result, we have incorporated the aforementioned experiments and the corresponding discussions regarding the STEM characterizations and analysis of PCCs in the revised manuscript (Pages 20-21) and Supplementary Information (Pages 102-104). The experimental procedures for ball-milling and surface acid treatment and MIL-101 were included in the “Methods” section in the manuscript.

Response to Comments of Reviewer #3:

General Comments:

The authors have addressed most of my concerns. But there are still some issues need revision.

Comment 1. It is highly recommended for the authors to provide high resolution TEM images and elemental mappings to directly confirm the introduction of PCC. Besides, no SEM images could be found in Figure 1.

Response:

We appreciate the valuable suggestion. In line with the comment made by Reviewer #2 (comment 2), we conducted STEM element linear scanning on four MOF@PCC samples, both with and without ball-milling treatment, to demonstrate the incorporation of PCCs on the MOF surface. The results of the STEM element linear scanning for all four MOF@PCC samples are illustrated in **Figure R1 & R2**. These images reveal a noticeable disparity in intensity between PCC and MOF, with a uniform distribution of signal intensity. By analyzing the MOF/PCC metal ratio along the cross-section of a single particle, we observed a volcano-shaped curve, indicating a higher concentration of MOF metal in the central region of the particle. This observation supports the notion that the PCC molecules undergo surface modification rather than being encapsulated within the MOF channels. Moreover, the ball-milling experiment and reversible PCC modification experiment on MIL-101 further suggested that the majority of PCCs are predominantly situated on the exterior surface of the MOF, with fewer PCCs encapsulation within the MOF particles, as evidenced by both the STEM images and analysis (**Figure R3-R7**).

Additionally, in the text explaining Figure 1, it should be “TEM” rather than “SEM”. We have made the necessary correction in the manuscript.

Figure R1. STEM images and linear scanning analysis with normalized intensities of (a) PCN-222@PCC-4, (b) PCN-222@PCC-5, (c) MIL-101@PCC-4, and (d) MIL-101@PCC-5.

Figure R2. Examinations of the (a) Zr/Co and (b) Zr/Pd element ratios on PCN-222@PCCs samples. The curves demonstrated a lower Zr/PCC ratio at the thin edge and a higher Zr/PCC ratio at the thick center.

Figure R3. STEM images, elemental mapping and linear scanning analysis of MIL-101@PCC-4 particles following various durations of ball-milling. (a) 0 h, (b) 1 h, (c) 2 h, (d) 3 h, and (e) 4 h.

Figure R4. STEM images, elemental mapping and linear scanning analysis of MIL-101@PCC-5 particles following various durations of ball-milling. (a) 0 h, (b) 1 h, (c) 2 h, (d) 3 h, and (e) 4 h.

Figure R5. Quantitative analysis of (a) the Co/Cr ratios and PCC-4 modification percentage of MIL-101@PCC-4 as well as (b) the Pd/Cr ratios and PCC-5 modification percentage of MIL-101@PCC-5 under ball-milling over time.

Figure R6. A schematic representation, STEM images, elemental mapping, and linear scanning analysis of the (a) PCN-222@PCC-4, (b) PCN-222@PCC-4 after acid treatment, and (c) reversible modification of PCC-5 on PCN-222.

Figure R7. STEM images and linear scanning analysis of the (a) PCN-222@PCC-4, (b) PCN-222@PCC-4 after acid treatment, and (c) reversible modification of PCC-5 on PCN-222.

Comment 2. It is very difficult to understand the so-called “hierarchical” nature of PCN-222 and MOF@PCC, the given N₂-sorption isotherms imply they are in microporous regime.

Response:

Thank you for pointing out this issue. To enhance clarity and avoid any potential misunderstandings, we have taken the reviewer's suggestion into account and removed the term “**hierarchical**” from the manuscript.

The corresponding changes have been highlighted in red in the manuscript.

Comment 3. The authors claimed they tried to emphasize the concept “using porous materials to modify porous MOFs”, then it may be inappropriate or unconvincing to employ a microporous PCC to modify the so-called “hierarchical PCN-222” (even if it is hierarchical porous) to obtain “enhanced properties”.

Response:

Thank you for the comments. We would like to clarify that our emphasis on the concept of “using porous materials to modify porous MOFs” stems from the fact that the “porous capping agents” are applied to the external surface of MOFs and this coating enhances the properties and performance of MOFs without compromising the transportation of guest molecules or the porosity of the parent framework. Our intention was not to imply that the porous capping agents contribute additional porosity to the parent MOF structure.

To ensure clarity and address any potential misunderstandings, we have removed the term “hierarchical” from the manuscript. Furthermore, we have expanded the discussion in the main text,

specifically on **Page 3**, to provide a more comprehensive explanation of the concept of “using porous materials to modify porous MOFs”.

REVIEWERS' COMMENTS

Reviewer #2 (Remarks to the Author):

The authors have addressed my comments, I think this manuscript could be published in the current version.

Reviewer #3 (Remarks to the Author):

The authors have addressed the reviewers' concerns and the revised version is recommended for publication.